# Biocompatible, Multi-Mode, Fluorescent, *T*_2_ MRI Contrast Magnetoelectric-Silica Nanoparticles (MagSiNs), for On-Demand Doxorubicin Delivery to Metastatic Cancer Cells

**DOI:** 10.3390/ph15101216

**Published:** 2022-09-30

**Authors:** Margo Waters, Juliane Hopf, Emma Tam, Stephanie Wallace, Jordan Chang, Zach Bennett, Hadrian Aquino, Ryan K. Roeder, Paul Helquist, M. Sharon Stack, Prakash D. Nallathamby

**Affiliations:** 1Department of Pre-Professional Studies, University of Notre Dame, Notre Dame, IN 46556, USA; 2The Berthiaume Institute for Precision Health, University of Notre Dame, Notre Dame, IN 46556, USA; 3Department of Art, Art History & Design, University of Notre Dame, Notre Dame, IN 46556, USA; 4Department of Mathematics and Pre-Professional Studies, University of Notre Dame, Notre Dame, IN 46556, USA; 5Department of Aerospace and Mechanical Engineering, University of Notre Dame, Notre Dame, IN 46556, USA; 6Department of Electrical Engineering, University of Notre Dame, Notre Dame, IN 46556, USA; 7Bioengineering Graduate Program in the Department of Aerospace and Mechanical Engineering, University of Notre Dame, Notre Dame, IN 46556, USA; 8Department of Chemistry and Biochemistry, University of Notre Dame, Notre Dame, IN 46556, USA; 9Harper Cancer Research Institute, University of Notre Dame, Notre Dame, IN 46556, USA

**Keywords:** magnetoelectric, *T*_2_-contrast, fluorescent nanoparticle, cobalt ferrite, silica shell, ON-Demand drug delivery, stimuli responsive drug delivery, doxorubicin, off-target toxicity

## Abstract

There is a need to improve current cancer treatment regimens to reduce systemic toxicity, to positively impact the quality-of-life post-treatment. We hypothesized the negation of off-target toxicity of anthracyclines (e.g., Doxorubicin) by delivering Doxorubicin on magneto-electric silica nanoparticles (Dox-MagSiNs) to cancer cells. Dox-MagSiNs were completely biocompatible with all cell types and are therapeutically inert till the release of Doxorubicin from the MagSiNs at the cancer cells location. The MagSiNs themselves are comprised of biocompatible components with a magnetostrictive cobalt ferrite core (4–6 nm) surrounded by a piezoelectric fused silica shell of 1.5 nm to 2 nm thickness. The MagSiNs possess *T*_2_-MRI contrast properties on par with RESOVIST™ due to their cobalt ferrite core. Additionally, the silica shell surrounding the core was volume loaded with green or red fluorophores to fluorescently track the MagSiNs in vitro. This makes the MagSiNs a suitable candidate for trackable, drug nanocarriers. We used metastatic triple-negative breast cancer cells (MDAMB231), ovarian cancer cells (A2780), and prostate cancer cells (PC3) as our model cancer cell lines. Human umbilical vein endothelial cells (HUVEC) were used as control cell lines to represent blood-vessel cells that suffer from the systemic toxicity of Doxorubicin. In the presence of an external magnetic field that is 300× times lower than an MRI field, we successfully nanoporated the cancer cells, then triggered the release of 500 nM of doxorubicin from Dox-MagSiNs to successfully kill >50% PC3, >50% A2780 cells, and killed 125% more MDAMB231 cells than free Dox.HCl. In control HUVECs, the Dox-MagSiNs did not nanoporate into the HUVECS and did not exhibited any cytotoxicity at all when there was no triggered release of Dox.HCl. Currently, the major advantages of our approach are, (i) the MagSiNs are biocompatible in vitro and in vivo; (ii) the label-free nanoporation of Dox-MagSiNs into cancer cells and not the model blood vessel cell line; (iii) the complete cancellation of the cytotoxicity of Doxorubicin in the Dox-MagSiNs form; (iv) the clinical impact of such a nanocarrier will be that it will be possible to increase the current upper limit for cumulative-dosages of anthracyclines through multiple dosing, which in turn will improve the anti-cancer efficacy of anthracyclines.

## 1. Introduction

The use of anthracyclines, such as Doxorubicin (Dox) in cancer treatment, is limited by a number of side effects, which include the acute reversible toxicities of nausea, vomiting, stomatitis, and bone marrow suppression [1,2]. The efficacy of anthracyclines in treating cancer is further limited by dose-dependent systemic toxicity (e.g., cardiotoxicity, neurotoxicity, vascular toxicity, etc.), with a cumulative dose > 550 mg/m^2^ causing an increase in the prevalence of heart failure and vascular damage [2]. This progressive toxicity usually manifests after anthracycline therapy and may become apparent within one year of the completion of treatment (early onset) or many years after chemotherapy has been completed (late-onset) [3]. The long-term organ toxicity caused by the anthracyclines includes for example vascular dysfunction, irreversible cardiomyocyte death and therefore chronic reduced heart function [3]. Recent studies of breast cancer survivors have also consistently shown changes in their cognitive function following chemotherapy, including memory loss, a tendency for lack of focus, and difficulty in performing simultaneous multiple tasks [3,4]. These cognitive problems, collectively called somnolence or cognitive dysfunction, are also reported in cancer patients, especially breast cancer patients, undergoing Dox-based chemotherapy. Despite their numerous side-effects, some of which are chronic, anthracyclines such as Dox, remain an important class of chemotherapeutic agents against solid-tumors, which makes abandoning them not an option [1,2,5,6].

Another factor affecting Anthracycline efficacy is that approximately 50% of Dox is eliminated from the body without any change in its structure, while the remainder of the drug is processed through three major metabolic pathways [7]. Metabolism of anthracyclines occurs through hydroxylation, semiquinone formation or deoxyaglycone formation, which can result in the formation of metabolites that either augment or suppress the anticancer properties of anthracyclines [8]. Consequently, localizing Dox specifically to cancer cells will increase exposure of cancer cells to a larger cumulative dose while negating the off-target metabolism and systemic toxicity of the drug.

Our in vitro results indicates that using magneto-electric silica nanoparticles (MagSiNs) as drug-delivery vehicles for Dox will avert off-target toxicity. Magneto-electric nanoparticles are heterostructures composed of a magnetostrictive core encased within a piezoelectric shell [8]. Magnetostriction is a reversible property of ferromagnetic materials (e.g., cobalt ferrite) which causes them to expand or contract in response to a magnetic field [9]. Piezoelectricity is the reversible appearance of a positive charge on one face and a negative charge on the opposite face (a voltage, in other words) of certain solid materials (e.g., fused silica, barium titanate) when they are subject to mechanical stress (by squeezing it) [10]. In a magneto-electric nanoparticle, application of a magnetic field will induce a change in the dimension of the magnetostrictive core which will transfer strain through the interface to the piezoelectric shell [11]. Consequently, a charge polarization and change in zeta (ζ) potential is introduced on the shell surface through the piezoelectric process, which is the desired magneto-electric effect.

We tested three different modes of cargo loading on the nanoparticles to determine the mode with the most stable payload and optimum release kinetics. In technique 1, we synthesized magnetoelectric nanoparticles from peer-reviewed publications (cobalt ferrite core with a barium titanate shell also known as MENs) and electrostatically loaded Fluorescein in the glycerol mono-oleate shell surrounding the Barium Titanate. In technique 2, we linked fluorescein to the MagSiNs surface (cobalt ferrite core with a fused silica shell) through variable length acid-sensitive ester linkers for pH dependent payload release. In technique 3, we covalently immobilized the fluorescein on the MagSiNs using CLICK chemistry followed by an external alternating magnetic field to trigger the release of the payload. The oft-cited magneto-electric nanoparticles (MENs) used as a drug nanocarrier possess a cobalt ferrite (CoFe_2_O_4_) core and a barium titanate shell [12,13,14,15]. However, the barium titanate shell was suitable only for electrostatic loading of drug molecules on the surface which resulted in significant leaching of drug molecules at a steady rate [13,14,15]. The barium titanate shell was uneven and increased the total diameter of the core–shell nanoparticles from 3–4 nm to 40–50 nm [12]. The 40–50 nm diameter reduced the surface area available per unit mass of nanoparticles by three orders of magnitude in comparison to 4–5 nm nanoparticles. So, for our study, we created magneto-electric nanoparticles with a cobalt ferrite core and a fused silica shell (Magneto-silica nanoparticles; MagSiNs) with net diameters in the range of 4–6 nm to increase surface area available for drug conjugation. The MagSiNs had *T*_2_-MRI contrast properties (negative contrast) because of the cobalt-ferrite core [16]. The MagSiNs were also trackable by fluorescence from the fluorophores volume-loaded in their silica shell [17,18]. Of the three loading techniques explored, technique 3 had zero leaching of the payload and the most favorable release kinetics of the payload as well (~100% payload released per hour). So we proceeded to use technique 3 to immobilize Dox. HCl on MagSiNs.

The silica shell on MagSiNs provides a highly stable substrate for silanization (-Si-O-Si- bond) [19] by silane derivatized doxorubicin hydrochloride (Dox-HCl) molecules. The acid anhydride on a silane is linked to the amine on Dox-HCl to form silane derivatized Dox-HCl. Additionally, in an alternating magnetic field generated by an electromagnet coupled to a sinusoidal alternating current (AC) generator, the cobalt ferrite core will undergo pulsed magnetostriction, leading to a stress-wave being pulsed through the thin piezoelectric fused silica shell [20,21]. The pulsed stress-wave destabilized the surface silane bonds between the silica shell and the silane-derivatized Dox, while the charge polarization due to the piezoelectric effect electrostatically repelled the negatively charged Dox-HCl from the MagSiNs surface, leading to active drug delivery. The CoFe_2_O_4_ core additionally gave the MagSiNs *T*_2_-weighted MRI contrast [16].

Transmission electron microscopy (TEM) data (Figure 1A indicated that the Dox-MagSiNs had an average diameter of 4.7 nm ± 2.5 nm, which included a piezoelectric shell thickness of 1.5 nm to 2 nm. Dox-MagSiNs possess silane derivatized Dox-HCl. ζ−Potential measurements of Dox-MagSiNs increased from −6.8 mV to −11.2 mV to −13.4 mV to −25.6 mV in the presence of magnetic field strengths of 0 Gauss, 30 Gauss, 40 Gauss and 265 Gauss which confirmed the magneto-electric nature of our nanocarriers similar to the existing literature. The Dox delivering capability of the MagSiNs was tested against three metastatic cancer cell lines: (i) A2780, an ovarian carcinoma; (ii) MDAMB231, triple-negative, metastatic, epithelial breast cancer cells; and (iii) PC3, metastatic epithelial derived prostate cancer cells. Intracellular co-localization experiments using lysosomes tracking dye showed localization of MagSiNs relative to the cell structures such as cell membrane, lysosomes, and the cytosol of cancer cells when a 30 Gauss magnetic field was applied. Existing, studies have shown that magnetic fields that are a least 100 times stronger and magnetic nanoparticles that are 100 nm to 500 nm in size are required for brute force permeation of cells [22]. Contrary to the existing literature that uses a 2 to 3-order of magnitude larger magnetic field (~3000–30,000 Gauss) [23,24] to permeabilize cells, our data indicates nano-poration of the cancer cell membrane by the magneto-electric MagSiNs into the cytosol, since the magnetic field of 30 Gauss is too weak to brute force nanoparticles into cells. After nano-poration of the cancer cells by the Dox-MagSiNs (500 nM w.r.t Dox-HCl), when an AC magnetic field of 30 Gauss at 100 Hz was applied for either 0.75 h, 1.5 h, 3 h or 10 h, this resulted in ~up to 60% apoptosis/necrosis of cancer cells and near 100% death of the control non-porated human umbilical vein endothelial cells (HUVECs). Crucially in the absence of an AC magnetic field, the Dox-MagSiNs were completely biocompatible with all cell types pointing to the stable nature of the silane bond of the Dox with the silica shell. This was the opposite result of both a 20 nM and a 500 nM dose of nanocarrier free Dox-HCl, which killed 100% of the HUVECs irrespective of the presence of the AC magnetic field. The stable loading of Dox-HCl on MagSiNs was crucial to avoid off-target systemic toxicity.

Our results indicated that off-target, systemic toxicity of anthracyclines (e.g., Dox) can be mitigated by delivering Dox on magneto-electric silica nanoparticles (Dox-MagSiNs) because the Dox-MagSiNs are therapeutically inert like pro-drugs. ON-Demand onco-toxicity was achieved once the Dox.HCl from Dox-MagSiNs were triggered released by an external alternating magnetic field.

## 2. Results and Discussion

### 2.1. Physical Characterization of MagSiNs

The MagSiNs nanoparticles made of a novel core–shell composition cobaltferrite@silica (CoFe_2_O_4_@SiO_2_), in which the relatively high moment CoFe_2_O was used to enhance the magneto-electric coefficient [25]. Previous studies [14] have focused on proof-of-concept experiments with no consideration for scaling-up manufacture or batch to batch consistency [15]. In this study, despite the novelty of our core–shell NPs for on-demand drug release, we ensured that our wet synthesis yielded NPs in the 0.1 kg range as opposed to current state of the art that yield milligrams of nanomaterials [26]. A typical transmission electron microscopy image of the fabricated CoFe_2_O_4_, is shown in Figure 1A(i,ii). Transmission electron microscopy images showing CoFe_2_O_4_ nanoparticle cores capped in a fused silica shell silica is shown in Figure 1A(iii,iv). The crystal lattice of CoFe2O_4_ nanoparticles was clearly visible at 150,000–200,000× magnifications, as was the silica shell. The silica-shell thickness was 1.51 ± 0.94 nm. The lattice spacing on the CoFe_2_O_4_ nanoparticle cores was 0.865 ± 0.038 nm which is consistent with cubic spinel crystal structure from the literature (Figure 1A(ii)) [27]. The MENs had barium titanate shell with shell thickness varying from 10 nm to 40 nm (Figure 1A(v,vi)). From the TEM image analysis the MagSiNs diameter was 6.71 ± 2.48 nm (Figure 1B(i)). From the TEM image analysis the MENs diameter was 37.1 ± 13.9 nm. (Figure 1B(ii)). The elemental composition of the MagSiNs was confirmed through energy-dispersive spectroscopy, to be Co, Fe, O, and Si as shown in Figure 1C(i). The elemental composition of the MENs was confirmed through energy-dispersive spectroscopy, to be Co, Fe, O, Ba, and Ti as shown in Figure 1C(ii). The EDXS peak heights are consistent with the stoichiometry of Co:Fe = 1:2.

The magnetic hysteresis loops of the sample cobalt ferrite NPs and MagSiNs were measured using a Microsense EV7 vibrating sample magnetometer (VSM) (Figure 2A,B). During the operation of the VSM, the CoFe_2_O_4_ or the MagSiNs sample were vibrated between pick up coils. A DC external field from an electromagnet magnetized the sample. The resulting oscillating magnetic field from the sample induced an alternating emf which was proportional to the total magnetic moment of the sample. Sweeping the external magnetic field and measuring the resulting magnetic moment gave the magnetic hysteresis loop of the sample. The nanoparticles were deposited on 1 × 1 cm^2^ silicon dies and measured in the VSM. The magnetic field was swept from −18 to 18 kGa. The contribution of the silicon and silicon dioxide was removed by removing the slope of the curve using the slope from 10 kGauss to 18 kGauss where the samples are saturated. The magnetic properties of the samples are summarized in Table 1 below. The saturation magnetization (emu/g) of MagSiNs was 53.20 emu/g (in-plane) and 3.70 emu/g (out-plane) in comparison to Cobalt Ferrite which was72.14 emu/g (in-plane) and 39.07 emu/g (out-plane). The saturation magnetization is the point at which the material obtains maximum alignment with the applied magnetic field. This corresponds to the maximum magnetic moment that can be obtained. The magnetostrictive coefficient is only applicable until the saturation magnetization because the material reaches its maximum strain and cannot continue to elongate at this point. The maximum elongation also corresponds to a 90 degree Bloch wall and domain rotation over the total volume of material. In other words, the saturation magnetization varies between materials and occurs when their domains have all been aligned. Any increase of the applied magnetic field past the saturation point would then not affect the material because its domains could not be further aligned. When the applied magnetic field is removed and no stress is applied to the material, the magnetostriction exhibits some hysteresis. This occurs because the material is still magnetized when the magnetic field is removed. which results in nonzero strain in the material. Magnetostrictive materials are used to convert electromagnetic energy into mechanical energy and vice versa. The magnetic field or force applied would create a strain in the material. The VSM hysteresis loop confirmed the suitability of the magnetostrictive properties of the cobalt ferrite core in the MagSiNs to produce vibrations in the presence of an alternating magnetic field, which can then induce charge polarization on the piezoelectric silica shell of MagSiNs. VSM measurements were indicative of favorable magnetostrictive properties that is crucial for the magnetoelectric effect and ON-Demand drug release from MagSiNs.

The MagSiNs size distribution fell within 4.3 nm to 9.1 nm with peak diameter centered around 6.7 nm. This size range is ideal for materials being designed with in vivo applications in mind. Nanomaterials that are >5 nm avoid being filtered out through the renal system [28,29]. The narrow size range between 4.3 nm to 9.1 nm size also makes it easier to model nanoparticle distribution in a flowing fluid. The >5 nm MagSiNs can also circulate multiple times through the blood circulatory system allowing higher probability of localizing to the target tissue.

### 2.2. Characterization of Drug Nanocarrier Properties

In Table 2, we explored the stability of the drug payload on the nanocarriers and the release kinetics of different stimuli responsive drug delivery mechanism. Several drug-loading mechanisms were explored. We tried to reproduce electrostatic loading and release of fluorescein isothiocyanate (FITC-a Dox proxy) from cobalt ferrite@barium titanate core–shell magneto-electric nanoparticles (MENs) (Appendix A) [30]. We also modified a previous technique of ours to encapsulate cobalt ferrite cores in a piezoelectric fused-silica shell (MagSiNs) [18,31]. We used ester linkers to immobilize FITC on MagSiNs to explore mimicking acid hydrolysis of ester bonds in lysosomes to release the active form of the drug from MagSiNs [32,33]. We used a 2-carbon long ethyl linker and a 4-carbon long butyl linker between the ester bond and FITC to see how linker length will affect FITC release. We also explored the immobilizing FITC on MagSiNs through the formation of an amide bond [34,35] with carboxylate groups on the surface of the MagSiNs followed by low frequency (50 Hz to 100 Hz) AC magnetic field induced release of FITC from MagSiNs.

The electrostatic loading of FITC on MENs was not stable and it leached the fluorescent payload constantly. 100% of the fluorescent payload was released in 1-h of a 100 Hz alternating magnetic field at 27–30 Gauss. However, 84% of the fluorescent payload was released in 1-h even without the magnetic field. Additionally, the barium titanate shell was not of reproducible thickness leading to a broad distribution in the size and shape of CoFe_2_O_3_-BaTiO_3_ core–shell NPs. These two reasons will lead to inconsistent drug loading, unpredictable drug release, and unwarranted off-target toxicity. For these reasons, we decided not to pursue the MENs architecture or electrostatic loading of drugs.

For our second technique we wanted to exploit the acidic environment of cancer cells to trigger the release of drugs from MagSiNs. It is well documented that by increasing the length of the carbon spacer between the ester bond and the drug molecules, it is possible to control the ester hydrolysis rate and thereby the drug release rate from the nanocarrier [36,37]. The ester hydrolysis rate is higher at acidic pH and therefore drug release is expected to accelerate in the acidic environs of lysosome-like organelles as well as acidic extra-cellular matrix of cancer cells. We tested the release of FITC at two pH values: pH 4.75 (MES buffer) and pH 7.2 (PBS buffer). The ester bond with the butyl linker released the payload at 0.98%/hour at pH 7.2 and at 2.1%/hour at pH 4.75. This translated to a cumulative payload release of ~ 50.3% ± 3% at pH 4.75 over a 24-h time-period as opposed to 23.6% ± 1.2% at pH 7.2 for 24 h. These results matched our expectations. However, for the ester bond with the ethyl linker to the payload there was no significant pH dependence. The ester bond with the ethyl linker released the payload at 2.5%/hour at pH 7.2 and at 2.8%/hour at pH 4.75. This translated to a cumulative payload release of ~66.2% ± 3.4% at pH 4.75 over a 24-h time-period as opposed to 59.9% ± 2.5% at pH 7.2 for 24 h.

In either case the first-order kinetics of drug release and the ensuing compartmentalization rate of the drugs to the cancer cell would have been too slow to overcome the chemoresistive mechanisms of the cancer cells. Additionally, similar to electrostatic loading, there was a steady leaching of the fluorescent payload into the solution from the nanocarrier even at neutral pH which again made it unsuitable to avoid off-target toxicity.

In contrast, when the FITC payload was linked to the MagSiNs by means of an amide linker, the total free payload observed in solution after 24 h was 3.6% which remained a constant over 4-days. In the presence of the 27–30 Gauss, 100 Hz, AC magnetic field, up to 80% of the payload was released 30-min post-exposure to the AC magnetic field, which is the much preferred near-instantaneous release of payload from the nanocarriers. 90% cumulative release of payload and ~100% cumulative release of payload was measured at 3-h and 8-h post AC-field exposure.

The near instantaneous release profile of the payload from the nanocarrier means that cancer cells will be exposed to the full dose of the drugs in a short-burst which is favorable for compartmentalization of drugs to the cancer cells such that the drugs can exert their anti-cancer effect before being neutralized by the chemoresistant mechanisms of cancer cells. The lack of leaching of the payload from the MagSiNs surface in the absence of an AC magnetic field bodes well for negating off-target toxicity of such cancer therapeutics such as Dox. It is for these reasons that we decided to test Dox.HCl covalently immobilized on MagSiNs followed by AC magnetic field release of Dox.HCl as a label-free, ON-Demand chemotherapeutic delivering nanocarrier.

### 2.3. Characterization of the Magnetoelectric Properties of MagSiNs

VSM measurements (Figure 2A,B) showed magnetization and demagnetization repeatedly for both cobalt ferrite nanoparticles and MagSiNs highlighting the magnetostrictive property of the cobalt ferrite cores which is necessary to induce the magneto-electric effect in the piezoelectric shell. We tested this by measuring the zeta-potential of the MagSiNs in the presence of different magnetic-field strengths. The zeta potential range of cancer cells listed in Table 2 from a literature survey was ~−14 mV to −25 mV (A2780), 12.8 ± 2.8 mV (PC3), and −15 to −18 mV (MDAMB231) [38,39,40]. For HUVECs, which are derived from human umbilical vein, the zeta potential has been measured at 12.8 ± 0.56 mV [41]. Although the zeta potential of the control cells (HUVECs) and the cancer cells (A2780, PC3, MDAMb231) fell within the same dynamic range, the stiffness of HUVEC cell membranes are 10-fold to 1000-fold higher than the cancer cell membranes which were highly pliant (Table 2). Interestingly the zeta potential of MagSiNs can be tuned from −6.8 mV to−25.6 mV by using a magnetic-field from 0 Gauss to 265 Gauss (Table 3). This is due to the magnetostriction of the core in the magnetic field, which in turn deforms the piezoelectric shell to varying degrees, resulting in an increase in charge presentation on the MagSiNs surface. The MagSiNS did exhibit low zeta potential values. These values led to flocculation (reversible) over time at high concentrations (1% *w*/*v*) of MagSiNs, but not aggregation (irreversible). The MagSiNs could always be resuspended by vortexing. However, the kinetics of flocculation are also influenced by the initial concentration of nanoparticles in solution [42]. Since typically for our cell or animal studies we utilized a 1000 fold to 10,000 fold less than 1% *w*/*v*, we did not have any issues with stability of dispersion. For the magnetic field of 27–30 Gauss the zeta-potential of the MagSiNs was between −11 to −15 mV. This pointed to the ability of the MagSiNs to match the membrane potential of cancer cell which in turn will enable them to interact with the cell membrane for prolonged durations without repulsion, and to actively electronanoporate across the cancer cell membranes [43]. This in theory will make MagSiNs ideal drug nanocarriers.

### 2.4. Characterization of Image Contrast Properties of MagSiNs

The magnetic resonance image (MRI) contrast enhancing efficacy of the synthesized spherical cobalt-ferrite and fluorescent MagSiNs nanostructures (*T*_2_ agent) is characterized by its relaxivity coefficient (*r*_2_), which is related to *T*_2_ through Equation (1) [44]
1/T_2_ = 1/T_2_^0^ + r_2_C (1)
where *C* is the contrast agent concentration, *T*_2_ is the observed relaxation time in the presence of cobalt ferrite nanostructures, and *T*_2_^0^ is the relaxation rate of pure water. In the equation, *T*_2_ becomes shorter when the concentration (*C*) increases, while *r*_2_ is the relaxivity coefficient. From the given equation, it reveals that as the concentration increases the 

MRI image appears darker and contrast agents having a higher *r*_2_ value require small concentration increments. In other words, unlike *T*_2_, which depends on concentration, *r*_2_ is a concentration-independent term. A contrast agent with a large *r*_2_ value can shorten *T*_2_ drastically with a smaller concentration increment. *T*_1_–*T*_2_ averaged MRI scans w.r.t the concentration of CoFe_2_O_4_ nanoparticles were taken. As expected CoFe_2_O_4_ nanoparticles showed a *T*_2_-weighted effect with concentration dependent enhancement of negative contrast in the image (Figure 2C). However, MagSiNs with the silica shell showed a *T*_2_-weighted, negative contrast effect only until 3 mM w.r.t CoFe_2_O_4_ concentration. However, it still demonstrated the efficacy of MagSiNs as MRI contrast agents (Figure 2D).

The efficacy of NPs as contrast agent for MRI is related to their relaxivity values (*r_1_, r_2_*). The ratios of relaxivities are reported with respect to the total molarity of iron and cobalt (i.e., s^−1^ mM^−1^ Fe) (Figure 2E). Ratio of transverse/longitudinal relaxivity (*r*_2_/*r*_1_) for iron-oxide NPs and commercially available *T*_2_-contrast agent RESOVIST [45] from were compared to the CoFe_2_O_4_. The cobalt ferrite NPs had *r*/*r*_2_ values of 26.9 ± 2.4 s^−1^ mM^−1^ which was comparable to the *r*_2_/*r*_1_ values of iron oxide NPs 28.3 ± 2.9 s^−1^ mM^−1^ [46]. However, the MagSiNs had an *r*_2_/*r*_1_ value that was significantly lower at 15.2 ± 1.7 s^−1^ mM^−1^ but still comparable to commercially available RESOVIST contrast agent at 17.4 ± 1.8 s^−1^ mM^−1^. This was interesting because CoFe_2_O_4_ nanoparticles showed a *T*_2_-weighted effect with higher negative contrast at increasing concentrations of CoFe_2_O_4_ up to 10 mM which was our highest concentration tested. MagSiNs showed *T*_2_-weighted image contrast up to 3 mM concentration w.r.t iron. However, at 10 mM MagSiNs exhibited *T*_1_-weighted image signal enhancement. This concentration dependent *T*_1_ or *T*_2_ enhancement by MagSiNs was very similar to published research with RESOVIST which initially showed *T*_1_-weighted signal enhancement immediately after administration to the patient but showed *T*_2_-weighted image contrast at approximately 10–15 min post administration and clearance [47].

After demonstrating the ability to reproducibly encapsulate the cobalt ferrite core in a silica shell and characterizing the magnetic and MRI properties of the core vs. the core–shell nanoparticles, we used optimized protocols previously published from our lab to incorporate fluorophores within the silica shell (volume-loading) to gain fluorescent modality without altering the surface properties of the MagSiNs [17,18,31,48,49,50]. Without covalent attachment, dye molecules weakly associated with the porous structure of the amorphous silica leak into the surrounding environment. This is the most common problem associated with the integration of organic dyes into silica nanoparticles. Many studies have attempted to resolve this problem by using coupling agents and chemical binding [18]. However, the low intensity in fluorescence and resulting low sensitivity of the organic dyes used limited their applications. By volume-loading the fluorophores into the silica shell, the fluorescence signal from fluorophores will also be impervious to solvent effects and pH effects. RITC or FITC which when linked to aminopropyltriethoxysilane were readily co-precipitated with tetraethoxysilane into the PVP mesh surrounding the cobalt ferrite nanoparticles to yield a discrete fluorescent silica shell by the modified Stöber method. This resulted in either red-fluorescent or green-fluorescent MagSiNs with steady fluorescent signals which enabled us to track the MagSiNs during in vitro studies (Figure 2F). We were therefore able to demonstrate the synthesis of MagSiNs with dual-modalities of detection (MRI and fluorescence). Fluorescence modality is suitable for tracking MagSiNs in vitro while MRI modality is suitable for tracking MagSiNs in vivo.

### 2.5. Characterization of Cytocompatibility, Biocompatibility and Biodistribution of MagSiNs

For suitability as a drug carrier, it was important to assess the cytocompatibility and biocompatibility of MagSiNs [51]. Cytocompatibility was assessed against HUVEC cells, which is model cell line for blood vessels and is utilized extensively to assess the cytocompatibility of intravenously delivered therapeutics. Biocompatibility was assessed in immunocompetent Balbc/J mice.

HUVEC cells were incubated with 0.116 µg MagSiNs for 48 h. A simple LIVE/Dead assay using calcein AM ester/propidium iodide was used to differentiate live cells from dead cells (Figure 3A). Statistically there was no difference in cell viability of control HUVECs grown in complete growth medium vs. HUVECs grown in media supplemented with MagSiNs. The control groups had 84 ± 10.5% live cells and viable cells in MagSiNs exposed test cell group had 71 ± 8.9% live cells.

For the biocompatibility assessment, Balbc/J mice (*n* = 4) were each injected with 10 mg/kg MagSiNs through the tail-vein. There was a mice cohort for each time point (1 h,4 h,8 h,24 h,48 h) and each cohort was sacked at that time point post-injection of MagSiNs. At the endpoint, the brain, heart, lungs, liver, kidney, spleen and fecal pellets were collected and fixed in 4% buffered para-formaldehyde for further analysis. At least 100 μL to 500 μL of blood was harvested per mouse by cardiac puncture and stored between 2–8 °C. The collected tissue were scored for inflammation by H&E staining of histology sections (Figure 3B). Additionally, a known mass of each tissue was digested and analyzed by ICP-OES for quantifying the biodistribution of MagSiNs longitudinally.

Histology scoring of tissue samples up to 48 h post-exposure to MagSiNs did not indicate any inflammation in comparison to the control mice cohort. Ex vivo MRI imaging were acquired using *T*_1_ and *T*_2_ scans on the Bruker desktop 1T MRI (Figure 4A). To determine if the biodistribution trended towards bioclearance or bioaccumulation, we looked at the ratio of the integrated image intensity of the *T*_2_-weighted scan of the control mice tissue against that of the integrated image intensity of the *T*_2_-weighted scan of the mice tissue from each timepoint cohort (Figure 4B). This ratio had an exponential negative slope indicating insignificant non-specific accumulation of MagSiNs in mouse tissue and organs. Interestingly the ICP-OES analysis of the fecal pellet showed a sinusoidal curve with peak MagSiNs at 8 h and 48 h which indicated clearance of the MagSiNs through the G-I tract (Figure 4C). ICP-OES of the fecal pellets determined the control mice (no MagSiNs exposure) had a baseline signal of 10.7 ± 2.0 Fe (ppb) per milligram of sample. There was a statistically significant increased amount of Fe in the fecal pellet of mice injected with MagSiNs at the 4 h (36.2% higher) and 24 h (50.4% higher) post-injection mark. This indicated that the MagSiNs are indeed being cleared out through the G-I tract and not accumulating in vivo.

Our in vitro and in vivo testing prove the cytocompatibility and biocompatibility of our MagSiNs. Our biodistribution studies utilize *T*_2_-weighted MRI scans and ICP-OES to demonstrate effective clearance of the MagSiNs through the G-I track without any non-specific accumulation in tissues over a 48-h period post-injection. The biodistribution results, combined with the in vitro and in vivo biocompatibility results confirmed the suitability of utilizing MagSiNs as drug nanocarriers.

### 2.6. Anti-Cancer Efficacy of Dox Released from Dox-MagSiNs

Schematic illustration of the addition of (a) free Dox, and (b) silanized-Dox conjugated to magneto-electric silica nanoparticles (MagSiNs) to normal and cancer cells (Figure 5). The Dox-MagSiNs were incubated with the cells while exposed to a permanent magnet, followed by drug-release in an alternating electromagnetic field.

Dox-MagSiNs were added to the metastatic cancer cells from ATCC (MDA-MB-231, PC3, A2780) or normal HUVEC cells, exposed to a unidirectional magnetic field (24–50 Gauss) for 24 h, and then Dox.HCl release was triggered in a 100 Hz alternating electromagnetic field of the same strength to demonstrate that the ON-Demand release of Dox.HCl activates its cytotoxic activity. Simultaneously, as a control group, the same cell lines were also treated with free drugs alone to compare the anti-cancer efficacy of free drug formulations (Dox.HCl) to drug formulations delivered on MagSiNs (Dox-MagSiNs).

The viability assays (Figure 6) were executed for 3 total sample groups: (a) free Dox, (b) Dox-MagSiNs, and (c) cell-culture media. The concentration of Dox conjugated to the Dox-MagSiNs were matched by the concentration of free drug formulations (20 nM Dox.HCl or 500 nM Dox.HCl). Viable cell populations of treated and untreated control cancer cell lines and treated and untreated normal cell lines were determined at the end of each run using Calcein Am/propidium iodide assay to determine total cells and viable cell populations. The calcein-AM assay is based on the conversion of the cell permeant non-fluorescent calcein AM dye to the fluorescent calcein dye by intracellular esterase activity in live cells. Propidium iodide (PI) is membrane impermeant and therefore does not enter viable cells with intact membranes. When PI does gain access to nucleic acids and intercalates its fluorescence increases dramatically and is therefore used to identify dead cells. Calcein-Am and propidium iodide (PI) can be used separately or together to assess cellular viability or cell death, respectively [52]. Statistical analysis using a paired *t*-test (α = 0.05) was used to confirm any significant difference in cell viability when comparing controls with cells exposed to Dox from Dox-MagSiNs.

In this study, for each sample set (control group or test group), we counted the total number of cells with green fluorescence and the total number of cells with red fluorescence as separate datasets. We determined that the green cells were representative of total cells in the image while the cells that emitted a red signal alone or red + green signal indicated the dead cells in the image. For each sample set we calculated the percent dead-cells and percent live-cells. We used the percent live-cells as a measure of viability. A paired *t*-test under the assumption of comparing two-samples with equal variances and α ≥ 0.05 was used to determine the statistical significance of the test-groups in comparison to the control-groups.

#### 2.6.1. HUVEC Control Cells (Figure 6A)

The control group for HUVECs consisted of three sample-sets. HUVECs were grow in cell culture medium, or were cultured in growth medium infused with MagSiNs equivalent to 500 nM Dox from Dox-MagSiNs. The control sample-sets cultured with MagSiNs were further split into two groups with one group exposed to no permanent magnetic field while the second group was exposed to 24 h of permanent magnetic field. Addition of free Dox.HCl at 20 nM and 500 nM resulted in 100% HUVEC death. However, when Dox-MagSiNs with 500 nM equivalent of Dox.HCl was incubated with HUVECs the viability of the exposed HUVECs was not statistically different from the control sample-sets. However, when either 20 nM or 500 nM equivalent dose of Dox release was triggered from Dox-MagSiNs using an AC magnetic field, this again resulted in 100% HUVEC cells death, similar to the free Dox.HCl doses.

The fact that for normal, control, HUVECs, 100% cell death was seen after exposure to 20 nM or 500 nM Dox.HCl in its free form or after release from MagSiNs indicates Dox activity is retained after release from Dox-MagSiNs. The most important result was the complete biocompatibility of Dox-MagSiNs to HUVECs in the absence of a AC magnetic field to trigger the release of the Dox.HCl from Dox-MagSiNs

#### 2.6.2. A2780 Ovarian Cancer Cells (Figure 6B)

The control group for A2780 consisted of three sample-sets similar to the HUVECS. The MagSiNs themselves were not toxic to the A2780 in the presence or absence of any magnetic fields. For metastatic ovarian cancer cells (A2780), 20 nM Dox in its free form killed > 20% cells. 20 nM Dox after release from MagSiNs resulted in no significant cell death. 83% cell death was seen after exposure to 500 nM Dox in its free form. 53% cell death was observed after 500 nM Dox was released from MagSiNs indicating reduced Dox anti-cancer efficacy after release from MagSiNs.

#### 2.6.3. PC3 Prostate Cancer Cells (Figure 6C)

The control group for PC3 consisted of three sample-sets similar to the HUVECS. For metastatic prostate cancer cells (PC3), 20 nM Dox in its free form or after release from Dox-MagSiNs resulted in >20% cell death. 100% cell death was seen after exposure to 500 nM Dox in its free form. 47% cell death was observed after 500 nM Dox was released from MagSiNs indicating reduced Dox anti-cancer efficacy after release from MagSiNs. The MagSiNs themselves were not toxic to the PC3 cells in the presence or absence of any magnetic fields. Interestingly, though 20 nM Dox.HCl in its free-form or when released from an equivalent dose of Dox-MagSiNs had similar anticancer activity. The similar anti-cancer efficacy of Dox.HCl from the two Dox.HCl formulations at the low dose (20 nM) but the dramatic difference in anti-cancer efficacy at the high dose (500 nM) might be indicative that the differences in the instantaneous dose of Dox.HCl released from Dox-MagSiNs becomes more pronounced at the higher dose [53]. The difference in instantaneous dose exposure is ~16 nM vs. 20 nM for the low dose and 400 nM vs. 500 nM for the higher dose, based on release kinetics from Table 1.

#### 2.6.4. MDAMB231 Triple-Negative Breast Cancer Cells (Figure 6D)

The control group for MDAMB231 consisted of three sample-sets similar to the HUVECS. The MagSiNs themselves were not toxic to the MDAMB231 cells in the presence or absence of any magnetic fields. For metastatic triple-negative breast cancer cells (MDAMB231), 20 nM Dox in its free form or after release from MagSiNs resulted in insignificant cell death. 4% cell death was seen after exposure to 500 nM Dox in its free form. ~10% cell death was observed after 500 nM Dox was released from MagSiNs indicating increased Dox anti-cancer efficacy after release from MagSiNs. However, overall MDAMB231 was significantly chemoresistant to the dosages of Dox.HCl that we administered in free-form or as Dox-MagSiNs. This was not surprising considering the highly efficient chemo-resistant mechanisms present in MDAMB231 [54].

We know from HUVECs that the Dox did not lose it activity after release from the Dox-MagSiNs. So this significant difference in anti-cancer activity of 500 nM Dox.HCl in its free-form and from Dox-MagSiNs might be due to instantaneous exposure of the cancer cells to the free form of Dox.HCl as opposed to the drug released from the Dox-MagSiNs which is 80% of the equivalent dose at 30 min post A.C. magnetic stimulation. However, the advantage here is that unlike standard chemotherapy, due to the non-cytotoxic nature of Dox-MagSiNs, there is the possibility of attacking the cancer cells with multiple doses of Dox-MagSiNs, which is not possible with free Dox.HCl due to the indiscriminate toxicity of Dox.HCl in its free form. Therefore, it is possible to increase the therapeutic window of standard chemotherapeutics like Dox.HCl by utilizing MagSiNs as drug carriers.

Overall, the MagSiNs themselves or 500 nM Dox-MagSiNs were not toxic to HUVEC cells or any of the cancer cells. HUVEC sensitivity to Dox induced toxicity is well known. Additionally, the membrane potential of HUVECs is also depolarized similar to cancer cells. However, since Dox from Dox-MagSiNs did not have enhanced anticancer efficacy against the cancer cells it is reasonable that the HUVEC cells death we saw was due to that cell lines increased sensitivity to Dox. HCl and not due to enhanced uptake of Dox-MagSiNs in a magnetic field. PC3 and A2780 showed statistically significant cell death after exposure to 500 nM Dox.HCl released from Dox-MagSiNs. 100% more MDAMB231 were killed with Dox. HCl released from Dox-MagSiNs in comparison to free Dox.HCl. However, there was still significant chemoresistant to Dox.HCl pointing to the need for a combinatorial treatment to nullify the chemoresistant mechanisms, to re-sensitize the MDAMB231 cells to Dox treatment. The biggest advantage of Dox-MagSiNs is that they negate non-specific toxicity from Dox.HCl, as was evident, with the 500 nM Dox-MagSiNs treated HUVECs having statistically similar viability to untreated HUVEC cells control groups. This implies that we can systemically deliver Dox-MagSiNs, still avoid off-target Dox.HCl toxicity, and deliver the full dose of Dox.HCl near instantaneously to the cancer by using a localized alternating magnetic field to trigger the release of the Dox.HCl. To elucidate the mode of interaction of the Dox.MagSiNs with the different cells we did co-localization studies by staining sub-cellular features and imaging using confocal microscopy.

### 2.7. Co-Localization Assays for Dox-MagSiNs in Cells

We carried out lysosomes co-localization assays similar to our previous publications [18,52,55] to elucidate a probable cause for the varied response of the different cancer cells to Dox.HCl released from MagSiNs. Co-localization assay for MagSiNs in cells was performed. The silica-shell of MagSiNs was volume loaded with Rhodamine-B red fluorescent dye. Lysosomes were stained with lysoview-green. Nucleus was stained blue with DAPI. The cell membrane was imaged suing phase-contrast illumination. Co-localization was assessed in the presence and in the absence of a 27–35 Gauss, permanent magnetic field.

#### 2.7.1. HUVEC Control Cells

Figure 7A showed no dependence on the external magnetic field on the co-localization of the MagSiNs extra-cellularly or intra-cellularly. Lysotracker dye indicated 3–5 lysosomes per cell. Co-localization study indicated that the number of MagSiNs clusters associated with HUVECs or internalized was similar regardless of the presence or absence of the weak magnetic field. The Pearson’s coefficient for co-localization with and without magnetic field were 0.87 and 0.74 respectively. HUVECs because of their stem-cell like origin from the umbilical veins, have depolarized membrane potentials ranging from −11 mV to −17 mV similar in range to the cancer cells tested here (Table 3). However, the Young’s modulus of their membrane (10–11 kPa) is 20-fold to 40-fold higher than the cancer cell lines tested here [56,57]. The co-localization study results indicate that the HUVEC cell death, when exposed to 20 nM or 500 nM Dox.HCl, either as free Dox.HCl or from Dox-MagSiNs, is due to the well documented sensitivity of HUVEC to Dox.HCl [58] and not because of enhanced uptake of Dox-MagSiNs.

#### 2.7.2. A2780 Ovarian Cancer Cells

Figure 7B had well defined and distinct lysosomes in the absence of a magnetic field. In the presence of a magnetic field, the MagSiNs were clustered to form 1–2 μm structures. Yellow fluorescence signal that indicated co-localization of MagSiNs with lysosomes accounted for less than 10% of the MagSiNs signal. Majority red fluorescence signal from MagSiNs indicated the lack of co-localization of the MagSiNs with lysosomes in the presence of a magnetic field. There was also a lack of distinct lysosomes in the cells in the presence of MagSiNs and a magnetic field. There is a distinct possibility that the >1 μm sized Dox-MagSiNs structures were able to disrupt lysosomes in the presence of the matching magnetic field [59],which would explain the lack of distinct lysosomes. The Pearson’s coefficient for co-localization with and without magnetic field were 0.89 and 0.92, respectively, which also concurred with the fact that there was only a slight increase in the co-localization signal between the green and red channel. While there was significant cell death in the presence of 500 nM dose of Dox.HCl as a free formulation or from 500 nM Dox-MagSiNs, the Dox delivered from Dox-MagSiNs resulted in 50–55% cancer cell death as opposed to 80–85% cell death observed with free Dox formulation. The fact that only 80% of Dox is released instantaneously from 500 nM Dox-MagSiNs, combined with about 10% of the Dox-MagSiNs being sequestered in lysosomes might account for the discrepancy in the efficacy of Dox released from Dox -MagSiNs in comparison to Dox used as a free formulation.

#### 2.7.3. PC3 Prostate Cancer Cells

Figure 7C had well defined and extensive lysosomes. As with A2780 we observed sequestration of the Dox-MagSiNs in lysosomes intracellularly in the absence of a magnetic field as evidenced by the dense overlays of the fluorescent signal of the MagSiNs with the lysosomes resulting in a strong yellow fluorescent signal. In the presence of a magnetic field, Dox-MagSiNs were clustered in cells, although they were not majority co-localized with the lysosomes and some well-delineated lysosomes remained. Approximately 19% of the fluorescent signal from Dox-MagSiNs was co-localized with the lysosomes. Furthermore, unlike A2780 cells, only ~8% of the Dox-MagSiNs clusters incubated with PC3 cells in the presence of a magnetic field were >1 μm. It’s possible that similar to A2780 cells, that MagSiNs in PC3 cells were sequestered into lysosomes, but that the >1 μm MagSiNs clusters were able to disrupt the lysosomes in the magnetic field similar to data from the existing peer-reviewed literature [59]. The Pearson’s coefficient for co-localization with and without magnetic field dropped from 0.95 to 0.86, which would support the theory that MagSiNs in a magnetic field can disrupt lysosomes. While there was significant cell death in the presence of 500 nM dose of Dox.HCl as a free formulation or from 500 nM Dox-MagSiNs, the Dox delivered from Dox-MagSiNs resulted in ~47% cancer cell death as opposed to >95% cell death observed with free Dox formulation. The reduced anticancer efficacy data of Dox.HCl released from Dox-MagSiNs correlates well with the co-localization data that indicated lysosomal sequestration of ~20% of the MagSiNs, which would imply that the instantaneous Dox.HCl dose released from Dox-MagSiNs will not be equivalent to 500 nM of free Dox.HCl.

#### 2.7.4. MDAMB231 Triple-Negative Breast Cancer Cells

Figure 7D were densely packed with lysosomes that agreed well with existing data from the literature [60]. In the absence of a magnetic field, the MagSiNs were barely co-localized with the cells (Pearson’ coefficient 0.77). In the presence of a magnetic field >85% of the MagSiNs fluorescent signals were heavily co-localized with the lysosomes in the overlay, indicating efficient sequestration of the MagSiNs in the lysosomes (Pearson’ coefficient 0.87). More than 70% of the >1 μm MagSiNs clusters were also co-localized with the lysosomes. While there appeared to be a slight increase of 10% cancer cell death in the presence of 500 nM Dox-MagSiNs in comparison to the 4% cancer cell death for a 500 nM dose of free Dox.HCl; unlike PC3 or A2780, it appears that even if the MagSiNs can efficiently disrupt the lysosomes in the presence of a magnetic field, there are enough number of lysosomes to re-sequester the MagSiNs. The co-localization study results would indicate the need for either using sufficiently large numbers of MagSiNs whose combined volume would overwhelm the combined volume capacity of lysosomes per cell or to modify MagSiNs with surface chemistry that would make them impossible to be sequestered in lysosomes. So although MagSiNs were efficiently internalized into MDAMB231, their sequestration in the lysosomes would indicate that the chemoresistant mechanisms of MDAMB231 were as efficient in negating the effect of 500 nM of free Dox as they were in negating Dox-released from Dox-MagSiNs based off the results of the cell viability studies [61,62].

## 3. Materials and Methods

### 3.1. Materials

Cobalt nitrate hexahydrate (Sigma-Aldrich, St. Louis, MO, USA ≥98%), iron nitrate nonahydrate (Sigma-Aldrich, St. Louis, MO, USA), polyvinylpyrrolidone (average molecular weight 40 kDA) (Sigma-Aldrich, St. Louis, MO, USA), sodium borohydride (Sigma-Aldrich, St. Louis, MO, USA 98%), de-ionized water (DI H_2_O), tetraethyl orthosilicate (TEOS) (Sigma-Aldrich, St. Louis, MO, USA ≥99.0% (GC)), 30% *w/v* ammonium hydroxide (NH_4_OH) (Sigma-Aldrich, St. Louis, MO, USA), 200 proof ethanol (VWR), fluorescein isothiocyanate isomer I (FITC) (Sigma-Aldrich, St. Louis, MO, USA ≥90%), rhodamine B isothiocyanate mixed isomers (RITC) (Sigma-Aldrich), amino propyl triethoxy silane (APTES) (Gelest, 97%), 3-triethoxysilylpropylsuccinic anhydride (SSA) (Gelest, Morrisville, PA, USA 95%), N-(3-Dimethylaminopropyl)-N′-ethylcarbodiimide hydrochloride (Sigma-Aldrich, ≥90%), doxorubicin hydrochloride (Dox-HCl) (Sigma-Aldrich, St. Louis, MO, USA, 98.0–102.0% (HPLC)), 4% buffered para-formaldehyde (VWR, Radnor, PA, USA), butanol amine (Sigma-Aldrich, St. Louis, MO, USA, 97%), and ethanolamine (Sigma-Aldrich, St. Louis, MO, USA, ≥98%). Dulbecco’s Modified Eagle’s Medium (DMEM), RPMI-1640, fetal bovine serum heat-inactivated (FBS), and trypsin-EDTA were purchased from Sigma-Aldrich, St. Louis, MO, USA. Complete endothelial cell growth medium was from R&D systems. MDAMB231, PC3, A2780, and HUVEC were from ATCC. Lysoview-green, and Calcein-AM/ethidium homodimer III LIVE/DEAD assay were purchased from Thermo Fisher Scientific (Waltham, MA, USA).

### 3.2. Nanoparticle Synthesis

#### 3.2.1. Synthesis of MagSiNs Core

The MagSiNs core (3–6 nm) was composed of CoFe_2_O_4_, and it was synthesized using the hydrothermal method. In beaker 1, 0.58 g of cobalt nitrate hexahydrate and 1.6 g iron nitrate nonahydrate were dissolved in 150 mL of deionized (DI) water. The contents of the beaker were stirred at 1000 rpm at 70 °C. In beaker 2, 2 g of polyvinylpyrrolidone (40,000 molecular weight) and 9 g of sodium borohydride were dissolved in 50 mL of DI water. The beaker 2 solution was then added dropwise into beaker 1, at 0.55 mL/minute. Once this addition had been completed, the temperature of the hotplate was increased to 90 °C, the stirring was decreased to 300 rpm, and the solution was left to sit while the water evaporated from the solution until the mass was very sticky and tar-like. Next, the overhead stir paddle was removed from the container, and 150 mL of DI water was added to the beaker, and the nanoparticle-core mass was sonicated in the ultrasound bath for at least 15 min. The CoFe_2_O_4_ nanoparticles were then magnetically separated from the supernatant. This process of rinsing the MagSiNs cores with DI water was repeated 3 more times followed by 3 rinses in ethanol. The cores are dried in an oven at 60 °C and stored as a powder at room-temperature until addition of the silica shell.

#### 3.2.2. Synthesis of Silica Shell on MagSiNs Cores

The materials need for the synthesis of the silica shell on the Mag-E-Si-N cores were 200 proof ethanol, tetraethyl orthosilicate (TEOS), 20% *w/v* ammonium hydroxide (NH_4_OH), a sonic dismembrator, a centrifuge, an overhead non-magnetic stirrer, and a 400 mL beaker. 57 mg of the nanoparticle-cores was weighed out and added to a 50 mL centrifuge tube. 10–20 mL of 200 proof ethanol are added, and the tube was then placed in the sonic dismembrator and sonicated for 60 s at 40% amplitude (pulse on for 1 s, pulse off for 0.5 s). The core solution was then transferred to the 400 mL beaker, and additional 200 proof ethanol was added to make the total volume of ethanol 99 mL. Next, 1.05 mL of TEOS was added, and the solution was sonicated again for 20 s at 40% amplitude. The beaker of the solution was then stirred with the overhead stirrer. While spinning, 3 mL of 30% *w/v* NH_4_OH was added. The container was sealed and the cores were left to spin for approximately 48 h. After 48 h, the core–shell MagSiNs were rinsed thrice in ethanol by centrifugation at 9000 rpm for 30 min each and finally stored at room temperature as a dried pellet. The MagSiNS were resuspended in between rinses by using sonication at 40–50% power settings.

#### 3.2.3. Synthesis of Fluorescent MagSiNs

Green fluorescence (FITC) or red fluorescence (RITC) incorporated MagSiNs were synthesized in a similar manner with minor modifications. The particles in 30 mL anhydrous ethanol were dried and 22.6 mg was weighed out and resuspended in 40 mL of 200 proof ethanol in a 45 mL centrifuge tube. The solution was sonicated at 60% amplitude for 30 s. After, 340 μL of tetraethyl orthosilicate APTES (TEOS) was added. The solution was transferred to a 100 mL flask and was then covered with aluminum foil before 92 μL of the RITC fluorophore was added. The beaker was taken to the fume hood and placed under an overhead stirrer on low spin speed. Finally, 875 μL of 30% ammonium hydroxide was added before parafilm was placed on the top of the flask to reduce ethanol evaporation and more aluminum foil was added to reduce light exposure to the fluorophore. The solution spun for 24 h before being washed 3 times in 35 mL 200 proof ethanol for 15 min in the centrifuge at 9000 rpm. After the final wash, the cores were resuspended in 30 mL ethanol and stored. The same process was conducted for the addition of the 92 μL FITC fluorophore to 22.4 mg of cores.

### 3.3. Nanoparticle Characterization

#### 3.3.1. Transmission Electron Microscope (TEM)

High-resolution transmission electron microscopy images (TEM) of silica capped cobalt-ferrite nanoparticles with a magnetic core and piezoelectric shell was carried out on a JEOL 2011 at 100 kV. 

#### 3.3.2. Vibrating Sample Magnetometer (VSM)

The total magnetic moment of cobalt ferrite nanoparticles and MagSiNs at saturation magnetic field strength were measured using a vibrating sample magnetometer (VSM) for in-plane and out-of-plane measurements. The VSM was measured using a Microsense EV7 VSM.

#### 3.3.3. Magnetic Resonance Imaging (MRI)

The longitudinal relaxation time (*r*_1_) and transverse relaxation time (*r*_2_) were determined for cobalt ferrite nanoparticles, and MagSiNs using a 1T Bruker Benchtop Icon magnetic-resonance imaging instrument (MRI) in order to assess their suitability as MRI image contrast agents. Magnetic resonance imaging was performed with a Bruker Icon 1T MRI scanner running Paravision 6.0.1 for preclinical MRI research. CoFe_2_O_4_ and MagSiNs were diluted to 1 mM, 3 mM, and 10 mM concentrations in deionized water. After a three-plane localizer scan, *T*_2_ relaxation time was acquired with an MSME sequence protocol (*T*_2_map-MSME). Echo time (TE) was varied from 18 to 198 with 18-ms increments with the following parameters: TR = 2500 ms, matrix = 192 × 192, FOV = 35 × 35 mm, resolution = 0.182 × 0.182 mm, bandwidth = 15,000 Hz, slice thickness = 1.250 mm, and total acquisition time = 8 min. *T*_1_-weighted MR images were acquired using a *T*_1_ Rapid Imaging with Refocused Echoes (RARE) sequence (*T*_1__RARE) at various repetition times (TR) under the following parameters: TE = 12.0 ms, TR = 161.4, 400, 700, 1000, 1300, 1600 ms, matrix = 128 × 128, FOV = 30.0 × 30.0 mm, resolution = 0.234 × 0.234, bandwidth = 12,500 Hz, slice thickness = 1 mm, and total acquisition time = 33 min. *T*_2_ relaxation time was measured after selecting a region-of-interest (ROI) from the generated *T*_2_ maps. Signal Intensity (SI) was measured with ROIs from the generated *T*_1_ images at various TRs. In Matlab (Mathworks), SI versus TR was plotted and a two-parameter fit using Mz(t) = Mo (1 – e^(−t/*T*_1_)) was performed to calculate *T*_1_. Further image analysis was performed with ImageJ. The *r*_1_ and *r*_2_ values were calculated by determining the slope of 1/*T*_1_ and 1/*T*_2_ (s^−1^) versus sample concentration (mM). 

#### 3.3.4. Fluorescence Microscopy

Fluorescence of the cores after the addition of the fluorophores was analyzed via epi-fluorescent microscopy (Nikon eclipse 400 Melville, NY, USA) using standard green (fluorescein) and red (rhodamine) filter cubes. 

#### 3.3.5. Zeta Potential Measurement in the Presence of Magnetic Field

The zeta potential of MagSiNs was measured to characterize the zeta potential with and without the influence of a magnetic. The magnetic field was applied perpendicular to the electrical field of the electrodes. 1 mL of DI water and 20 μL of the MagSiNs was used as a solution. The refractive index of silica was used to calibrate the light scattering measurements. DLS and Zeta potential were measured at room temperature. Deionized water was the solvent. Each DLS or Zeta potential measurement file consisted of 3 runs. Each run was from an average of at least 15 measurement readings The zeta potential was measured using a Malvern Panalytical Zetasizer Nano ZS/ZSE.

### 3.4. Linking Drug-Proxy (FITC) to MagSiNs

#### 3.4.1. MagSiNs Surface Functionalized with FITC (FITC-MagSiNs)

Aminopropyltriethoxysilane was reacted with FITC (APTES-FITC) in 1:1 mole ratio, using THF as a solvent under nitrogen atmosphere. The reaction was allowed to proceed for 24 h under room temperature. The solution was then stored in −20 °C. MagSiNs and APTES-FITC were mixed in 1:10 mole ratio in a 90/10 ethanol/water solution and allowed to stir at room temperature for 24 h. After 24 h, the silanized nanoparticles (MagSiNs-SSA) were magnetically separated from solution, rinsed in DI water twice, and resuspended in DI water. The amount of fluorophore on the -FITC-MagSiNs will be quantified using fluorescence spectroscopy and comparison of fluorescence signal to a standard fluorescence calibration curve.

#### 3.4.2. MagSiNs Surface Functionalized with FITC through an Ethyl Ester Linker (MagSiNs-ethyl-FITC)

MagSiNs were functionalized with acid anhydride group using silanization with 3-(triethoxysilyl) propylsuccinic anhydride (SSA) in 90/10 Ethanol/water solution. Silanization was carried out for 24 h at room temperature. After 24 h, the silanized nanoparticles (MagSiNs-SSA) were magnetically separated from solution, rinsed in DI water twice, and resuspended in DI water. The acid anhydride groups on MagSiNs-SSA were reacted with 1000-fold mole excess of rhodamine tagged ethanolamine (ethanolamine-FITC) for 24 h at 4 °C. After 24 h, the silanized nanoparticles (MagSiNs-SSA) were magnetically separated from solution, rinsed in DI water twice, lyophilized and stored at −20 °C. The amount of fluorophore on the MagSiNs-ethyl-FITC will be quantified using fluorescence spectroscopy and comparison of fluorescence signal to a standard fluorescence calibration curve.

#### 3.4.3. MagSiNs Surface Functionalized with FITC through a Butyl Ester Linker (MagSiNs-butyl-FITC)

MagSiNs were functionalized with FITC through a 4-carbon linker (butanolamine-FITC) using the same experimental workflow as that of MagSiNs surface functionalized with FITC through an ethyl ester linker. The amount of fluorophore on the MagSiNs-butyl-FITC will be quantified using fluorescence spectroscopy and comparison of fluorescence signal to a standard fluorescence calibration curve.

### 3.5. Kinetics of Drug-Proxy (FITC) Release from MagSiNS for Different Payload Release Mechanisms

#### 3.5.1. ON-Demand FITC Release

An alternating magnetic field of 100 Hz with a field strength in the range of 27–35 Gauss was applied to vials of MagSiNs-FITC in phosphate-buffered saline, in a 5% CO_2_ cell-incubator at 37 °C. Vials were removed at 0.5 h, 1 h, 1.5 h, 3 h, 8 h; the nanoparticles were spun out, and the supernatant’s fluorescence signals were measured. The amount of fluorophore in the supernatant was quantified by comparing the fluorescence intensity to a standard calibration curve (Appendix A).

#### 3.5.2. Acid-Labile Ester Hydrolysis Dependent FITC Release

MagSiNs-ethyl-FITC (Appendix A) and MagSiNs-butyl-FITC (Appendix A) were suspended in vials of either phosphate-buffered saline at pH 7.2 or MES buffer at pH 4.75. The sample vials in pH 7.2 of 4.75 were placed in a 5% CO_2_ cell-incubator at 37 °C. The vials were sampled at 24 h, 48 h, 72 h, and 96 h; the nanoparticles were spun out, and the supernatant’s fluorescence signals were measured. The amount of fluorophore in the supernatant was quantified by comparing the fluorescence intensity to a standard calibration curve (Appendix A).

### 3.6. Linking Doxorubicin Hydrochloride (Dox.HCl) to MagSiNs (Dox-MagSiNs)

The MagSiNS were silanized with succinic acid. Typically 100 μL of (3-Triethoxysilyl) propylsuccinic anhydride, 95% was dispersed in 5 mL of a 90/10 (*v*/*v*) mixture of 200 proof ethanol and deionized (DI) water and then added to 5 mg of MagSiNs. The mixture was allowed to stir overnight. The silanized MagSiNs were then magnetically separated from solution 3X, and rinsed in DI water. We used Zeta potential measurements using Malvern panalytical Zetasizer Nano ZS/ZSE to confirm the presence of succinic acid anhydride (SSA) on the surface of MagSiNs by monitoring the dramatic change in zeta potential between SSA functionalized MagSiNs (−16.13 ± 0.76 mV) vs. non-functionalized MagSiNs (−6.8 mV).

Dox.HCl has an amine group on the cyclo-hexane group present in its structure. This amine group was reacted with the acid anhydride on MagSiNs-SSA in sterile DI water in the presence of 100 μg of EDC hydrochloride. The reaction was carried at 4 °C for 24 h, after which the Dox functionalized nanoparticles (Dox-MagSiNs) were magnetically separated from solution, rinsed in DI water twice, lyophilized and stored at −20 °C. Doxorubicin.Hydrochloride (Dox.HCl) is red in color with a distinct UV-Vis spectrum in the visible range [63]. The absorbance maximum of Dox.HCl is 480 nm. A calibration curve for known concentration of Dox.HCl was constructed using the absorbance max at 480 nm. After conjugating known mass of Dox.HCl to the SSA functionalized MagSiNs, we spun the MagSiNs out using centrifugation, and analyzed the supernatant for mass of unbound Dox.HCl. From the unbound mass of Dox.HCl we were able to determine the amount of Dox.HCl that was loaded on the NPs. The amount of Dox.HCl loaded on the MagSiNs-Dox was determined using UV-Vis spectroscopy to determine absorbance at 480 nm and comparing that absorbance to a standard calibration curve for Dox.HCl.

### 3.7. Magnetic Field Exposures

Electromagnet array used for exposing cells to unidirectional magnetic field by passing a DC current or an alternating magnetic field by passing an AC current through the electromagnet array. The electromagnet array consisted of 2 × 6 (12 in total) electromagnets with each row of electromagnets connected in series. The electromagnet array was made in house (Appendix A). A Hewlett-Packard S33120A waveform generator (Appendix A) and a Krohn-Hite Model 7500 Wideband Power Amplifier 115/230 V 50–400 Hz (Appendix A) were used to ensure that the electromagnets exhibited a magnetic field of between 23 Gauss to 40 Gauss. A DC current was applied to generate a unidirectional magnetic field. An AC current with a square waveform and 100 Hz frequency was used to generate an alternating magnetic field to trigger drug release.

### 3.8. Cell Culture

#### 3.8.1. Cell Stock

For the in vitro part of the experiment, a water bath, centrifuge, and a tissue culture hood were used, along with nutrient-rich media. Three cancer cell lines were cultured—MDAMB231 (breast cancer), A2780 (ovarian cancer), and PC3 (prostate cancer). One control cell line, human umbilical vein endothelial cell (HUVEC), was also cultured. The cells were cultured in a 5% CO_2_ incubator at 100% humidity. With regard to the nutrient-rich media, the type varied with each cell line. DMEM plus 10% FBS was used for MDAMB231. RPMI-1640 plus 10% FBS and 1% L-Glutamine was used for A2780 and PC3. Endothelial Cell Growth Medium (R&D systems) was used for the HUVEC cell line.

#### 3.8.2. Seeding of Experimental Chamber-Slides

To seed the cells, 8-well chamber slides, glass cover slide, and a hemocytometer are needed. The cells were ideally at 70% to 80% confluence before seeding. 15,000 to 20,000 cells were seeded into each well of the chamber slides, along with 400 μL of complete cell growth media appropriate for each cell line.

### 3.9. Cell Assays

#### 3.9.1. Cell Viability Assay

The cell viability assay consisted of seeding of cells into chamber slides, adding Dox (free or as Dox-MagSiNs), exposing slides to a permanent magnet for 24 h, exposing to an alternating current (AC) magnetic field for 10 h, and then performing a live/dead assay on the cells. The media in each well of the chamber slides is carefully aspirated, and the wells were rinsed twice with 400 μL of 1× PBS. The cells were incubated with ethidium-homodimer III dye and calcein-AM dye mix at manufacturer recommended concentrations (Life-technologies), and the slides were incubated for 30 min in the cell incubator, in the dark. After incubation, the wells were rinsed twice with 400 µL of 1× PBS. Lastly, 300 µL of 4% buffered para-formaldehyde were added to each well, and the slides were taken to be imaged with a Nikon inverted fluorescent microscope with the 10X lens and using a standard FITC/RITC green and red fluorescent filter cubes.

#### 3.9.2. Intra-Cellular Co-Localization Assay

HUVEC, A2780, MDAMB231, PC3 cells were plated onto sterile coverslips and allowed to adhere for 24 h; incubated with red-fluorescent MagSiNs equivalent to the dose of Dox-MagSiNs that would deliver 20 nM Dox.HCl; and exposed 24 h to a magnetic field (27–35 Gauss). The medium was then replaced, and the cells were incubated with the nanoparticles for the time indicated. To track the endocytic pathway, the cells were labeled 30 min with Lysoview DND Green 50 nM (Life Technologies Ltd., Carlsbad, CA, USA.). The wells are rinsed twice with 400 µL of 1× PBS. Lastly, cells were fixed in 4% buffered para-formaldehyde, stained with DAPI and the slides were taken to be imaged with a Nikon fluorescent microscope with the 10X lens and the green, blue and red fluorescence channels.

### 3.10. Animal Experiments

#### 3.10.1. Biodistribution

Balbc/J mice were purchased from Jackson Labs (ME). Five time cohorts (1 h, 4 h, 8 h, 24 h, 48 h post-MagSiNs injection) (*n* = 3–4/cohort) were created. The mice were restrained and 200 μL of 10 mg/mL MagSiNs in sterile 1× PBS was injected through the tail vein. The mice were ~ 7 weeks old. The control mice had only sterile 1× PBS injected into them. At each time point the mice were sacked by anesthetizing them with isofluorane followed by cervical dislocation. The blood was drawn by cardiac puncture. The brain, heart, lungs, liver, kidney, spleen, and fecal pellets in the large intestine were harvested and fixed in 4% buffered para-formaldehyde. The samples were all stored at 4 °C. All procedures were approved by the Institutional Animal Care and Use Committee at the University of Notre Dame and were conducted in accordance with the guidelines of the U.S. Public Health Service Policy for Humane Care and Use of Laboratory Animals.

#### 3.10.2. Inductively Coupled Plasma-Optical Emission Spectroscopy (ICP-OES)

Selected organs (liver, spleen, kidney, lung, heart, brain, intestine, and skin, blood), were dissected from three mice in each group, dried overnight in an oven at 37 °C, massed, and digested in aqua regia (3 HCl:1 HNO_3_) for 24 h. The mass of Fe, Co, Si in each sample was measured using ICPOES (Optima 8000, Perkin Elmer, Waltham, MA, USA). Calibration curves were created by diluting certified standard Fe, Co, and Si solutions (VWR, Radnor, PA, USA).

#### 3.10.3. Histology

The fixed organs were sliced, then the slice was rinsed with PBS, dehydrated in a graded series of ethanol solutions, embedded in paraffin, sectioned to 4 μm, and stained with hematoxylin and eosin. Stained tissue sections were imaged by transmitted light microscopy (Eclipse ME600, Nikon Instruments, Melville, NY, USA) at 1000× magnification and interpreted by a medical pathologist.

## 4. Conclusions

The difference in membrane physical properties of cancer cells [64,65,66] was exploited for chemotherapeutics delivery by Dox-MagSiNs by applying a magneto-electric charge and force that is above the threshold required to nanoporate the abnormal cells, but below the threshold required to nanoporate normal cells. We tested three different techniques to load the Dox.HCl on MagSiNs and determined that covalent immobilization of Dox.HCl on MagSiNs formed the most stable nanocarriers with near zero-order drug release kinetics. We have determined that an external magnetic field in the range of 25–50 Gauss at the cell membrane interface generates a magneto-electric charge and force on the Dox-MagSiNs that allows it to permeabilize the cancer cells and not the healthy cells (HUVECs). Preliminary studies confirmed that Dox was released on-demand from Dox-MagSiNs using an alternating magnetic field of 25–50 Gauss with a frequency range of 50 Hz-100 Hz which then proceeded to significantly kill (~50%) two out of the three cancer cells (A2780, PC3) we tested against, in a dose-dependent manner. When the Dox.HCl was released from the Dox-MagSiNs, the preferential accumulation of Dox-MagSiNs in cancer cells enhanced anti-cancer activity of Dox.HCl against two (PC3, A2780) out of the three cancer cell lines tested. The enhanced sensitivity of neo-vasculature such as those associated with cancer cells to existing chemotherapeutics resulted in 100% cell death of HUVECS once Dox.HCl was released from Dox-MagSiNs. This was despite the fact that there was no significant internalization of the Dox-MagSiNs by HUVECs in the low magnetic-field. Dox-MagSiNs also killed 50% more triple-negative breast cancer MDAMB231 cells in comparison to the same dosage of free Dox.HCl.

The three most important outcomes of our study were that (a) the drug-carrying magneto-electric nanocarriers (Dox-MagSiNs) are completely biocompatible, (b) a localized alternating magnetic field can be used to release the Dox.HCl from the Dox-MagSiNs in the vicinity of the tumor to negate any off-target toxicity associated with systemic delivery of drug molecules; (c) the rapid release kinetics of the payload from MagSiNs in the presence of an external alternating magnetic field ensures >50% cancer cell killing efficacy. Our concept is novel as we are tuning the surface electrical potential (the zeta-potential) of the MagSiNs to match the membrane potential of cancer cells to increase interaction with the cells and to ensure selective nanoporation into pliant cancer cell membranes and not the order-of-magnitude stiffer healthy cell membranes. While nanoporation into cancer cells resulted in >50% cell death of PC3 and >47% cell death of A2780 cells at the high-dose (500 nM) of Dox.HCl, toxicity from the Dox.HCl released from Dox-MagSiNs also resulted in HUVEC cell death. The ability of the Dox-MagSiNs in a localized alternating magnetic field to destroy cancer cells and associated tumor vasculature, is promising as this can lead to enhanced permeability and retention (EPR) effect. The MRI contrast properties along with fluorescence signal from our MagSiNs will also aid in image-guided localization to the tumor. It is also advantageous that we do not need to tag the Dox-MagSiNs with targeting molecules as some of the targeting labels are known to be systemically toxic themselves (e.g., Trastuzumab) [67]. Finally, since the Dox-MagSiNs are bio-inert, we can expose the cancer cells to multiple doses leading to higher cumulative doses of Dox-MagSiNs in comparison to free Dox.HCl. The systemic biocompatibility is a major step forward in increasing the tolerable total dose of chemotherapeutic that is currently allowed, thus increasing the therapeutic window and anti-cancer efficacy of an important class of existing chemotherapeutics such as anthracyclines, while essentially eliminating their harmful side-effects due to off-target activity.

## Figures and Tables

**Figure 1 pharmaceuticals-15-01216-f001:**
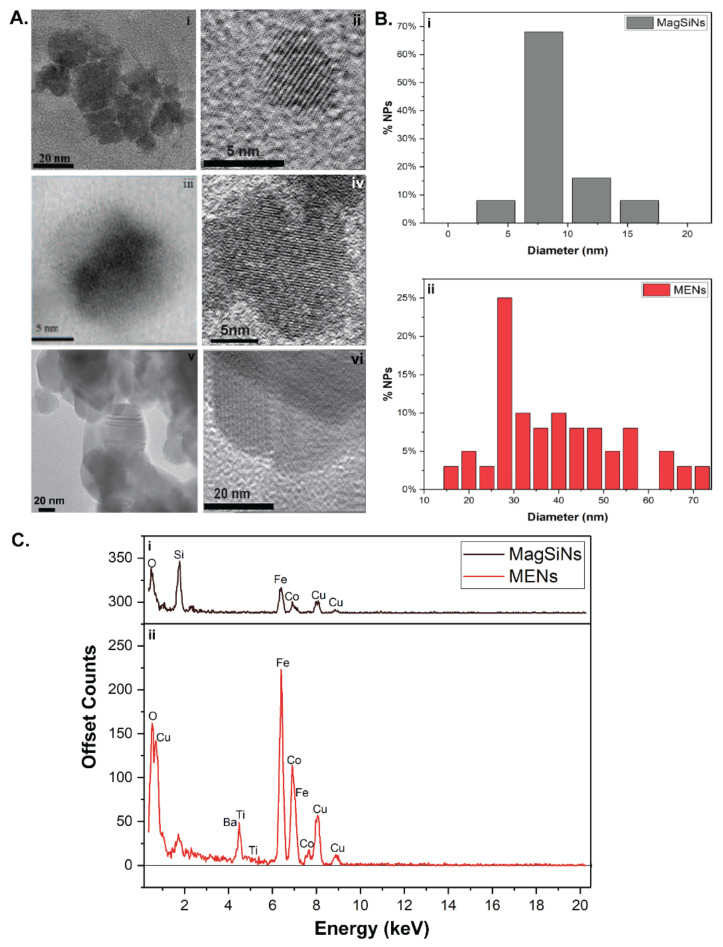
(**A**) Transmission electron microscopy images showing CoFe_2_O_4_ nanoparticle cores (**i**,**ii**), Magnetoelectric silica nanoparticles (MagSiNs) (**iii**,**iv**) and Magnetoelectric nanoparticles MENs (**v**,**vi**). The crystal lattice of CoFe_2_O_4_ nanoparticles was clearly visible in the higher magnifications. The lattice spacing on the CoFe_2_O_4_ nanoparticle cores was 0.865 ± 0.038 nm. (**B**) From the TEM images the MagSiNs average diameter was 6.7 ± 2.5 nm and MENs average diameter was 37.1 ± 13.9 nm. The silica-shell thickness was 1.51 ± 0.94 nm. The Barium Titanate shell thickness and shape was highly variable. (**C**) Energy dispersive X-ray spectrum (EDXS) of MagSiNs and MENs confirmed their distinct elemental composition differences in their shells. Distinct energy peaks for Fe (k_α_: 6.398 keV), Co (k_α_: 6.924 keV), Si (k_α_: 1.739 keV), and O (k_α_: 0.525 keV), were detected and labelled on MagSiNs samples. Distinct energy peaks for Fe (k_α_: 6.398 keV), Co (k_α_: 6.924 keV), Ba (L_α_: 4.465 keV), Ti (k_α_: 4.508 keV) and O (k_α_: 0.525 keV), were detected and labelled on MENs samples. The EDXS peak heights are consistent with the stoichiometry of Co: Fe = 1:2.

**Figure 2 pharmaceuticals-15-01216-f002:**
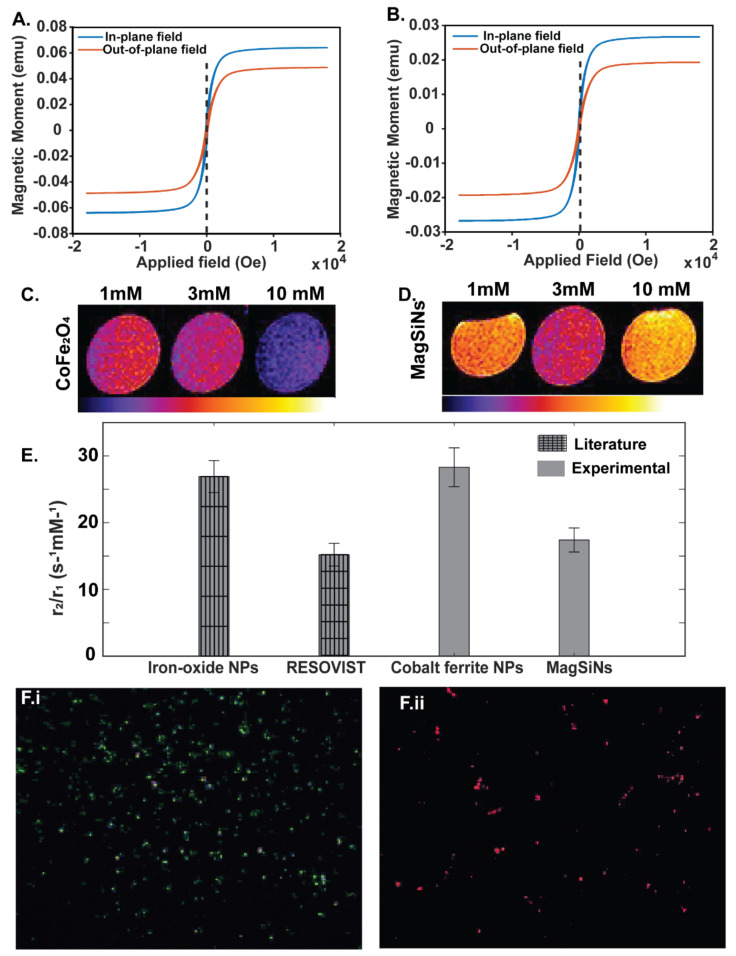
Dual-mode detection of MagSiNs. Vibrating sample magnetometry was used to measure magnetization as a function of magnetic field of (**A**) CoFe_2_O_4_ nanoparticles, and (**B**) MagSiNs in a reversible magnetic field at 300 K. *T*_1_ and *T*_2_ averaged MRI scans w.r.t the concentration of CoFe_2_O_4_ nanoparticles were taken. (**C**) As expected CoFe_2_O_4_ nanoparticles showed a *T*_2_-weighted effect with higher negative contrast at higher concentrations. (**D**) MagSiNs with the silica shell showed a *T*_2_-weighted, negative contrast effect only till 3 mM. However, it still demonstrated the efficacy of MagSiNs as MRI contrast agents. (**E**) Ratio of transverse/longitudinal relaxivity (*r*_2_/*r*_1_) for iron-oxide NPs and commercially available *T*_2_ contrast agent RESOVIST when compared to the CoFe_2_O_4_ nanoparticles and MagSiNs. MRI measurements were carried out in a 1T benchtop MRI. (**F**) Fluorophores embedded in the silica shell of MagSiNs to impart (**i**) green fluorescence from FITC or (**ii**) red fluorescence from RITC.

**Figure 3 pharmaceuticals-15-01216-f003:**
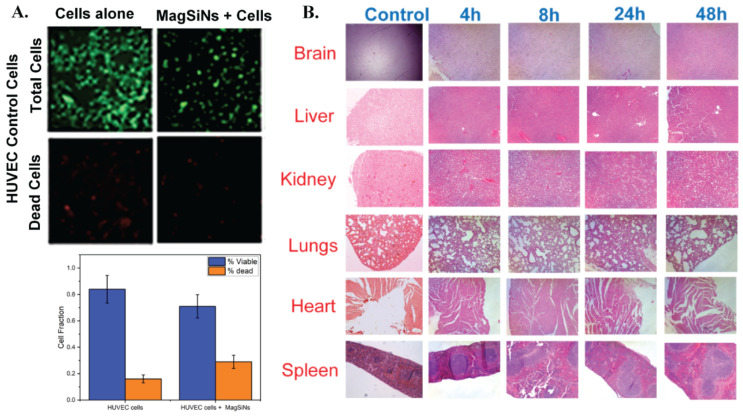
Assessing the cytocompatibility, and biocompatibility of MagSiNs to determine the usefulness of MagSiNs as drug nanocarriers. (**A**) The cytocompatibility of 0.116 micrograms of MagSiNs was assessed using the blood-vessel model cell line HUVECs. LIVE/Dead assay indicated that viable cells in control cells groups (84 ± 10.5%) and viable cells in MagSiNs exposed test cell group (71 ± 8.9%) had similar cell viability, 48 h post-exposure. (**B**) Biocompatibility was assessed using 4-time cohorts of BalbcJ mice (4 h, 8 h, 24 h, 48 h). There were 3 mice per cohort and the mice in each cohort were sacked at the pre-determined time after the 10 mg/kg MagSiNs injection. The brain, lungs, heart, liver, spleen, kidney, blood, and fecal pellets from the intestines were extracted and fixed in 4% buffered para-formaldehyde. Histology was scored for inflammation by a board-certified pathologist. There was no inflammation in any of the cohorts. The MagSiNs were cytocompatible, and biocompatible.

**Figure 4 pharmaceuticals-15-01216-f004:**
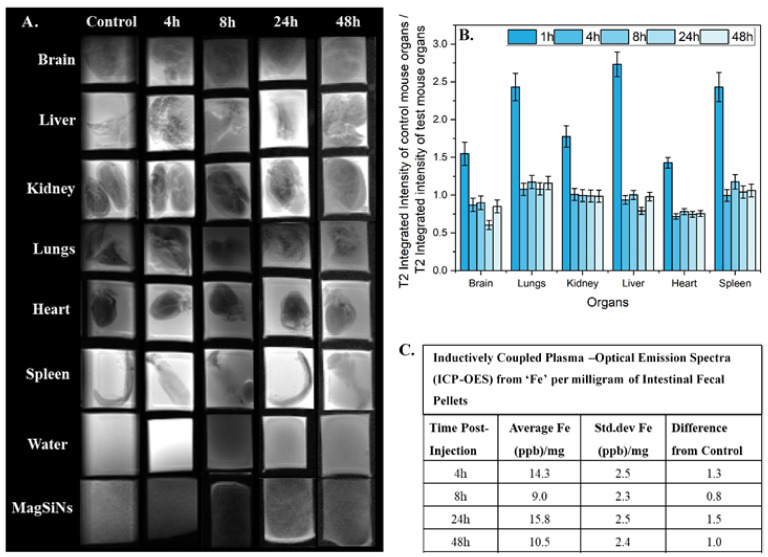
Assessing the biodistribution of MagSiNs to determine the usefulness of MagSiNs as drug nanocarriers. (**A**) Representative ex vivo *T*_2_-weighted, negative contrast, MRI scans of the mouse organs from the different time cohorts using the Bruker 1T benchtop MRI. The *T*_2_-weighted images were used to quantify the integrated intensity of the mouse organs from each time cohort. Di water and 10 mg/mL MagSiNs were used as control samples. (**B**) The ratio of the *T*_2_-weighted, integrated intensity of the signal from control mouse organs to that of the *T*_2_-weighted, integrated intensity of the signal from MagSiNs injected mouse organs was used to determine biodistribution kinetics over 48 h. There was no non-specific accumulation of the MagSiNs in any organs. (**C**) ICP-OES of the fecal pellets determined the control mice (no MagSiNs exposure) had a baseline signal of 10.7 ± 2.0 Fe (ppb) per milligram of sample. There was a statistically significant increased amount of Fe in the fecal pellet of mice injected with MagSiNs at the 4 h (36.2% higher) and 24 h (50.4% higher) post-injection mark. This indicated that the MagSiNs are indeed being cleared out through the G-I tract and not accumulating in vivo. The MagSiNs exhibited *T*_2_-MRI contrast and possessed favorable biodistribution, which made them suitable as drug nanocarriers.

**Figure 5 pharmaceuticals-15-01216-f005:**
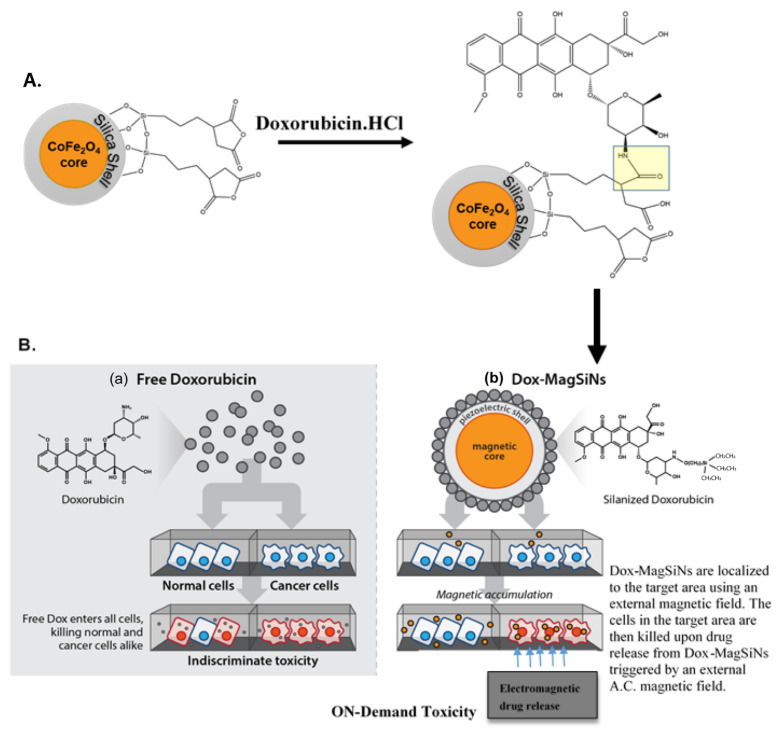
Schematic illustration of (**A**) the conjugation of doxorubicin to succinic acid anhydride group on MagSiNs through the formation of an amide bond between the amine group of the doxorubicin hydrochloride and the acid anhydride group on the MagSiNs. (**B**) Schematic of the experimental workflow of the addition of (**a**) free Dox.HCl, and (**b**) silanized-Dox.HCl conjugated to magneto-electric silica nanoparticles (MagSiNs) to normal and cancer cells. These drugs were then conjugated to the MagSiNs and incubated with the cells while exposed to a permanent magnet, followed by drug-release in an electromagnetic field. Illustration are not to scale. The highlight of this was the complete lack of cytotoxicity of the Dox-MagSiNs till the Dox.HCl was released from the Dox-MagSiNs by means of an externally applied electromagnetic field (30–50 Gauss).

**Figure 6 pharmaceuticals-15-01216-f006:**
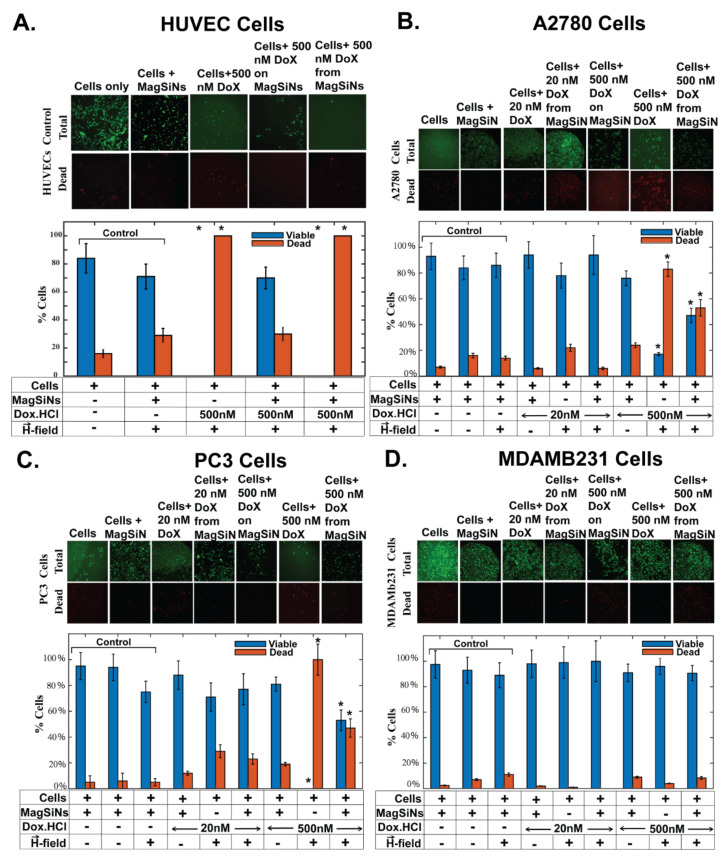
Total cells (green channel) versus dead cells (red channel) for all four cell lines in chamber slides exposed to Doxorubicin (Dox.HCl) released from Dox-MagSiNs compared to appropriate control groups. All the treated cells and controls presented here were exposed to a permanent magnet for 24 h, and then an alternating current magnetic field (A.C. mag field) of the same strength for 10 h. The live/dead assay was performed 48 h after initial drug exposure. (**A**) For normal, control, Human Umbilical Vein Endothelial Cells (HUVEC), 100% cell death was seen after exposure to 500 nM Dox.HCl in its free form or after release from MagSiNs indicating Dox.HCl activity is not lost after immobilization on MagSiNs; (**B**) for metastatic ovarian cancer cells (A2780), significant (83%) cell death was seen after exposure to 500 nM Dox.HCl in its free form. 53% cell death was observed after 500 nM Dox.HCl was released from Dox-MagSiNs which was still significant against control untreated cells. (**C**) for metastatic prostate cancer cells (PC3), 100% cell death was seen after exposure to 500 nM Dox.HCl in its free form. 47% cell death was observed after 500 nM Dox.HCl was released from MagSiNs; and (**D**) for metastatic triple-negative breast cancer cells (MDAMB231), 4% cell death was seen after exposure to 500 nM Dox.HCl in its free form. ~10% cell death was observed after 500 nM Dox.HCl was released from Dox-MagSiNs indicating increased Dox.HCl anti-cancer efficacy after release from Dox-MagSiNs. MagSiNs themselves or 500 nM Dox.HCl on Dox-MagSiNs were not toxic to HUVEC cells or any of the cancer cells.In the table below the bar graphs, ‘+’ indicates the presence of the component. ‘-’ indicates the absence of the component.

**Figure 7 pharmaceuticals-15-01216-f007:**
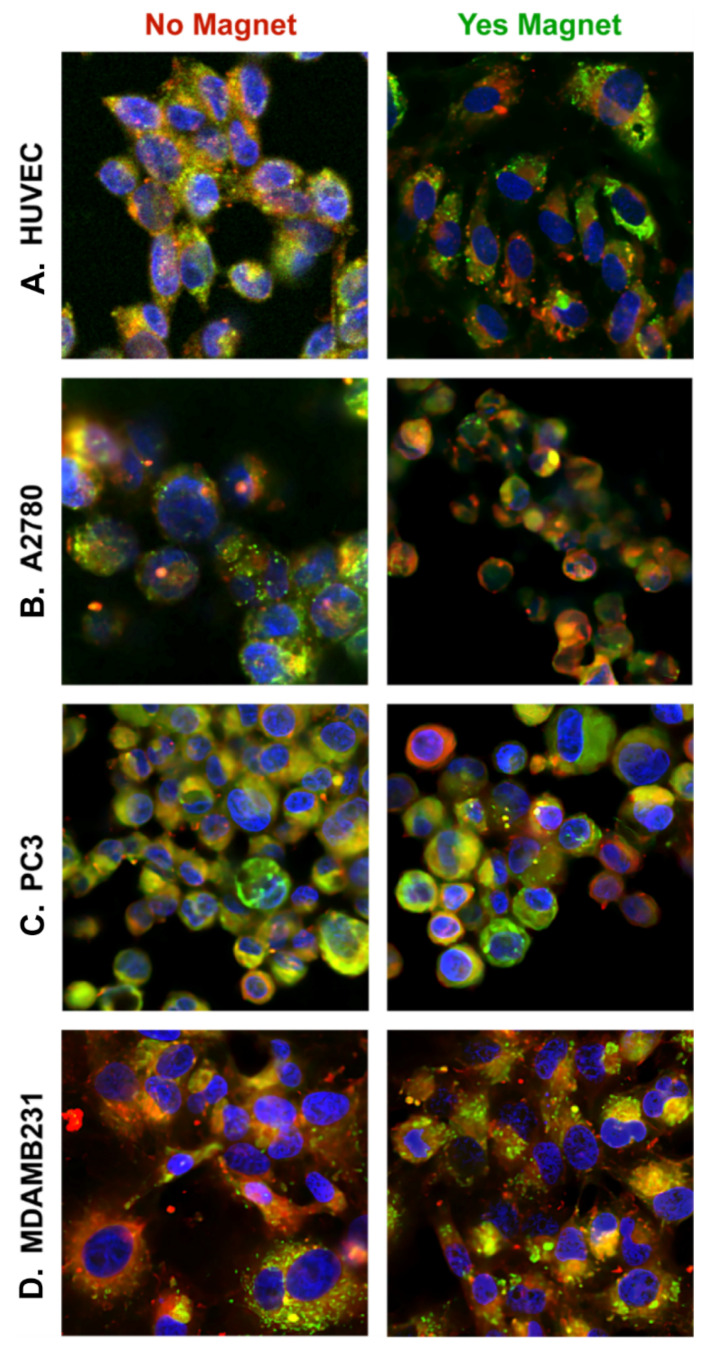
Co-localization assay for MagSiNs in cells was performed. The silica-shell of MagSiNs was volume loaded with Rhodamine-B red fluorescent dye. Lysosomes were stained with lysoview-green. Nucleus was stained with DAPI. Co-localization was assessed in the presence and in the absence of a 27–35 Gauss, permanent magnetic field. (**A**) HUVEC cells showed no dependence on the external magnetic field on the co-localization of the MagSiNs extra-cellularly or intra-cellularly. Pearson’s coefficient for co-localization with and without magnetic field were 0.87 and 0.74 respectively. (**B**) A2780 cells had well defined lysosomes in the absence of a magnetic field. In the presence of a magnetic field, the MagSiNs were clustered and seemed to be co-localized with lysosomes in the overlay. Pearson’s coefficient for co-localization with and without magnetic field were 0.89 and 0.92 respectively. (**C**) PC3 cells had well defined lysosomes. MagSiNs were co-localized in the lysosomes in the absence of a magnetic field (Pearson co-localization coefficient 0.95). In the presence of a magnetic field the majority of MagSiNs were not co-localized with the lysosomes even though a smaller population of well-delineated lysosomes remained. The Pearson co-localization coefficient dropped to 0.86. (**D**) MDAMB231 cells were densely packed with lysosomes. In the absence of a magnetic field, the MagSiNs were barely co-localized with the cells (Pearson co-localization coefficient 0.77). In the presence of a magnetic field the MagSiNs were heavily co-localized with the cells and especially with the lysosomes in the overlay, indicating efficient sequestration of the MagSiNs in the lysosomes (Pearson co-localization coefficient 0.87).

**Table 1 pharmaceuticals-15-01216-t001:** Comparison of Magnetic Properties of Cobalt-Ferrite Cores and MagSiNs from VSM measurements.

Sample	In-Plane or Out-of-Plane	Coercivity (Gauss)	Total Magnetic Moment at Saturation (emu)	Mass (g)	Saturation Magnetization (emu/g)
Cobalt Ferrite Cores	In-plane	120.68	1.01 × 10^−1^	1.40 × 10^−3^	72.14
	Out-of-plane	112.396	5.47 × 10^−2^	1.40 × 10^−3^	39.07
MagSiNs	In-plane	125.564	2.66 × 10^−2^	5.00 × 10^−4^	53.20
	Out-of-plane	119.742	1.85 × 10^−2^	5.00 × 10^−3^	3.70

**Table 2 pharmaceuticals-15-01216-t002:** Kinetics of payload release as a function of drug-release mechanisms. The payload RBITC was used as a proxy for drug molecules.

Nanoparticle Type	Linker Chemistry	100 Hz Magnetic Field Strength (Gauss)	pH	Observation Time Point (h)	% Payload (FITC) Released	Rate of Payload (FITC) Released
Drug Loading through Covalent Bond						
MagSiNs (Cobalt ferrite core and piezoelectric silica shell)	Isothiocyanate to amine	27–35	7.2	0.5 h magnetic stimulation	79.9% ± 0.2%	159.8% per hour
		27–35	7.2	1 h magnetic stimulation	78.9% ± 0.1%	78.9% per hour
		27–35	7.2	1.5 h magnetic stimulation	79.9% ± 0.2%	53.3% per hour
		27–35	7.2	3 h magnetic stimulation	90.1% ± 0.3%	30% per hour
		27–35	7.2	8 h magnetic stimulation	100% ± 0.3%	12.5% per hour
		0 (control)	7.2	48 h	3.6% ± 1.2%	0.075% per hour
Drug Loading through Covalent Ester Bond that is acid labile						
MagSiNs (Cobalt ferrite core and piezoelectric silica shell)	Ethyl-acetate ester bond	No magnetic field	7.2	24 h	59.9% ± 2.5%	2.5% per hour
		No magnetic field	4.75	24 h	66.2% ± 3.4%	2.8% per hour
MagSiNs (Cobalt ferrite core and piezoelectric silica shell)	Butyl-acetate ester bond	No magnetic field	7.2	24 h	23.6% ± 1.2%	0.98% per hour
		No magnetic field	4.75	24 h	50.3% ± 3%	2.1% per hour
Drug Loading through Hydrophobic Encapsulation						
MENs (Cobalt ferrite core and piezoelectric barium titanate shell)	Electrostatic loading in glycerol mono-oleate layer on MENs	27–35	7.2	1 h magnetic stimulation	100% ± 1.5%	100% per hour
		0	7.2	1 h	84% ± 2.1%	84% per hour

**Table 3 pharmaceuticals-15-01216-t003:** Magnetic field dependent zeta potential on the surface of MagSiNs.

**MagSiNs**	**Magnetic Field Strength (Gauss)**	**^₸^ Zeta Potential (mV)**	**Cells**	**Cell Lines**	*** Zeta Potential (mV)**	**^¥^ Cell Stiffness (kPa)**
	0	−6.8 ± 0.66		HUVEC	−12.8 ± 0.56	10–11
	30	−11.2 ± 1.5		A2780	−14 to −25	1.25 ± 0.5
	40	−13.4 ± 2.1		PC3	−12.8 ± 2.8	0.4 ± 0.05
	265	−25.6 ± 3.0		MDAMB231	−15 to −18	0.018 to 0.04

^₸^ Experimentally determined in our laboratory. * Cell zeta potentials determined from the peer-reviewed literature survey. ^¥^ Cell stiffness determined from the peer-reviewed literature survey.

## Data Availability

Data is contained within the article and Appendix A.

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
