# Peer review of "Biocompatible, Multi-Mode, Fluorescent, T2 MRI Contrast Magnetoelectric-Silica Nanoparticles (MagSiNs), for On-Demand Doxorubicin Delivery to Metastatic Cancer Cells"

_pharmaceuticals, 2022, doi:10.3390/ph15101216_

Round 1
Reviewer 1 Report
The current work focuses on Biocompatible, multi-mode, fluorescent, T2 MRI Contrast Magnetoelectric-Silica Nanoparticles (MagSiNs), for On-Demand Doxorubicin Delivery to Metastatic Cancer Cells. The author’s great effort into the manuscript, but minor issues should be addressed.
Title
The title is very long; make it shorter e.g. MagSiNs remove it
Abstract
- The abstract is very long, rephrase it with novelty and focus on the main outcomes that the manuscript will deal with it.
-make one system style 4-6 nm, 1.5-2 nm
Keywords
Insert keywords about your material, “Magnetoelectric-Silica Nanoparticles”
Introduction
-The introduction provides sufficient background, and all relevant references are included.
- The novelty of this work is not highlighted and it was not clear the author's contribution in comparison to other previous works.
- Separation between introduction section and results section: don’t make one mixed section.
- Line 106-107 need details and an explanation of why? “The MagSiNs had T2-MRI contrast properties (negative contrast) because of the cobalt-ferrite core”
Materials and Methods
- The used materials with their impurities should be inserted.
- What is the model version of each instrument? What is the country manufacture?
- What is the condition of measurements?
- Line 780, stir bar!! How use a magnet stirrer in the synthesis of magnetic nanoparticles? It will attract to it without a homogenous solution and product.
- Line 789, ammonium hydroxide (expired) (NH4OH)!
Result and discussion
-XRD analysis is required to detect the type of the magnetic phase with indexed peaks
- Elemental analysis and elemental distribution and mapping, which necessary to relate the efficiency of the material in terms of element percentage and distribution
- IR is required to confirm the organic layer e.g. silane layer and the presence of inorganic elements e.g. Fe, or linked to drugs
- Fig.4, needs correction
- One of the main problems in the manuscript is that the authors show only results without interpretations of it or confirmation by citation. More details are required to explain the obtained results.
Line 126-129, “ζ−Potential measurements of Dox-MagSiNs increased from -6.8mV to -11.2mV to -13.4mV to -25.6mV in the presence of magnetic field strengths of 0 Gauss, 30 Gauss, 40 Gauss, and 265 Gauss which confirmed the magneto-electric nature of our nanocarriers similar to existing literature.”
A zeta potential (ζ) is significant for biomedical applications to achieve expectable and consistent outcomes. The low value of zeta potential implies that the particle may show poor stability in aqueous solutions. Low zeta potential values (0 to ± 5 mV) will improve Van der Waals interparticle attractions and causes rapid coagulation and flocculation of particles. On the other hand, the higher value of zeta potentials implies that the particle may show good stability in aqueous solutions. There is a specific zeta potential value (≈ ± 30 mV) that determines the stability of particles. At this value, high electrostatic repulsive forces between the particles occur.
In this work, no interpretation of Zeta potential results!! What we can conclude from these numbers!! It's good for bio application or not? Why the sign was negative?
In addition, the condition of measurements is very important to judge the DLS and zeta potential results: What is the condition for measurement of zeta potential analysis? Concentration? pH? Solvent used type? One run or multi-run measurements?
- Figures TEM, improved are required to be clear with high resolution and quality. The core size and shell are not clear. In addition, the morphology was not homogenous
- In figure 1c (IV) what should we conclude from this picture: the indexed diffraction peak? Estimated size?
- Line 172 and 179, correct typo, CoFe2O
- Zoom in figure around zero for VSM analysis to clear coercive and Mr
- In Fig 2E, iron-oxide NPs and commercially available T2-contrast agent RESOVIST from literature when compared! Which literature? No information about the source of this figure!
- Most figures need more details in the text while the captions are supported with more detail
- All tables are needed to be re-organized with margin
Conclusion
The conclusion is very long; make it shorter e.g. Line 697-707, general information, removes it
References
- Correct the reference with complete citation e.g. no.51, 53, 62.
- make one system style and insert the more recent related citation.
Author Response
We thank the reviewer for their excellent and valuable feedback. Please see the attachment for our response.

Reviewer 2 Report
Overall comment:
Overall, the author formulated a magneto-electric silica nanoparticles (Dox-MagSiNs), serving as a controllable carrier for the deliver and release of Doxorubicin to cancer cells. With many different control tests, the results show that Dox-MagSiNs were completely biocompatible with all cell types and are therapeutically inert till the release of Doxorubicin from the MagSiNs at the cancer cells location. The author used metastatic triple-negative breast cancer cells (MDAMB231), ovarian cancer cells (A2780), and prostate cancer cells (PC3) as the model cancer cell lines. Human umbilical vein endothelial cells (HUVEC) were used as control cell lines to represent blood-vessel cells that suffer from the systemic toxicity of Doxorubicin. In the presence of an external magnetic field that is 300x times lower than an MRI field. Successful nanoporation of the cancer cells and triggered release of 500 nM of doxorubicin from Dox-MagSiNs to are observed. In control HUVECs, the Dox-MagSiNs did not nanoporate into the HUVECS and did not exhibit any cytotoxicity at all when there was no triggered release of Dox.HCl. The major advantages of this new nanoparticles formation are, (i) the MagSiNs are biocompatible in vitro and in vivo; (ii) the label-free nanoporation of Dox-MagSiNs into cancer cells and not the model blood vessel cell line; (iii) the complete cancellation of the cytotoxicity of Doxorubicin in the Dox-MagSiNs form; (iv) the clinical impact of such a nanocarrier will be that it will be possible to increase the current upper limit for cumulative-dosages of anthracyclines through multiple dosing, which in turn will improve the anticancer efficacy of anthracyclines.
Comment 1: The whole paper is well written and formulated with extensive work done, minor modification of the writing is needed.
Comment 2: Figure 4 and Figure 6, the labels should be revised.
Comment 3: In Figure 4 C, how is the difference from the control in the last column is calculated? It is a bit confusing.
Author Response
Comment 1: The whole paper is well written and formulated with extensive work done, minor modification of the writing is needed.
- We profusely thank the reviewer for their time and invaluable comments
Comment 2: Figure 4 and Figure 6, the labels should be revised.
- We have revised the labels as the reviewer has requested in our word document. We will make final edits to the Figures in the MDPI journal template once the reviewers approve the other coorections.
Comment 3: In Figure 4 C, how is the difference from the control in the last column is calculated? It is a bit confusing.
- We thank the reviewer for pointing this out and we have modified the table and figure caption. The control sample is a constant and acts as a baseline for Fe concentration. Fe in fecal pellets can only be present in mice which have been injected with MagSiNs, due to the Cobalt Ferrite core of MagSiNs. We compared the Fe signal at different timepoints from the fecal pellets of mice injected with MagSiNs against the baseline signal of Fe from the fecal pellets of control mice, which were not injected with MagSiNs. From this we deduced elevated Fe in fecal pellets 4h post MagSiNs injection (36.2% higher) and 24h post MagSiNs injections (50.4% higher). We have included this write up in the Results and Discussion section and the figure caption.